# IceR improves proteome coverage and data completeness in global and single-cell proteomics

Mathias Kalxdorf [1,2✉], Torsten Müller[1,3], Oliver Stegle [1,2] & Jeroen Krijgsveld [1,3✉]

Label-free proteomics by data-dependent acquisition enables the unbiased quantification of thousands of proteins, however it notoriously suffers from high rates of missing values, thus prohibiting consistent protein quantification across large sample cohorts. To solve this, we here present IceR (Ion current extraction Re-quantification), an efficient and user-friendly quantification workflow that combines high identification rates of data-dependent acquisition with low missing value rates similar to data-independent acquisition. Specifically, IceR uses ion current information for a hybrid peptide identification propagation approach with superior quantification precision, accuracy, reliability and data completeness compared to other quantitative workflows. Applied to plasma and single-cell proteomics data, IceR enhanced the number of reliably quantified proteins, improved discriminability between single-cell populations, and allowed reconstruction of a developmental trajectory. IceR will be useful to improve performance of large scale global as well as low-input proteomics applications, facilitated by its availability as an easy-to-use R-package.

[1] German Cancer Research Center, Heidelberg, Germany. [2] Genome Biology Unit, European Molecular Biology Laboratory, Heidelberg, Germany. [3] Heidelberg University, Medical Faculty, Heidelberg, Germany. ✉email: mathiaskalxdorf@gmail.com; j.krijgsveld@dkfz-heidelberg.de

The reproducible quantification of peptides and proteins across large sample cohorts is crucial to investigate proteome differences between individuals and across conditions[1]. Mass spectrometry is the leading technology to achieve this, operating to identify and quantify peptides that are generated from cell or tissue lysates, often in conjunction with liquid chromatography to maximize the number of detected peptides and thus enhance sampling depth[2]. The two main experimental approaches operate either via data-dependent or data-independent acquisition (DDA and DIA, respectively)[3]. In the traditional and more commonly used DDA approach, a predefined number of most abundant precursor ions (Top N) detected in an MS1 survey scan are sequentially selected for MS2-based peptide fragmentation and protein identification[4]. Although this has been successfully used to characterize thousands of proteins in countless proteomic studies, many peptides escape fragmentation due to the stochasticity of precursor selection, thus leading to missing data[5,6]. This is caused by proteome complexity and dynamic range, and it persists despite continuous improvements of sensitivity and acquisition speeds of mass spectrometers[7]. The degree of missing data becomes increasingly prominent in complex samples and in large sample cohorts[8], resulting in decreased numbers of proteins that can be consistently identified across all specimens[9], and hence limiting the power of DDA in large-scale proteomics, e.g. for biomarker discovery. To reduce missing value rates in DDA, several experimental and computational strategies have been developed in the last years. First, DIA co-fragments peptides in bins across the entire mass range, resulting in considerable advancements in reproducible protein measurement[10,11]. However, DIA suffers from the limited depth of quantifications due to the burden of peptide identification from severely convoluted data[8,12], and it requires an upfront reference library typically acquired by DDA[13]. Second, multiplexed labelling strategies such as TMT allow the simultaneous detection and quantification of peptides in an 11 or even 16-plexed manner at significantly reduced rates of missing values[14]. This approach has also been used for single-cell analysis, using one of the TMT channels to boost the signal in MS1 to benefit detection of peptides in the other channels[15,16]. Yet, the optimal size of the booster channel as well as quantification of low-abundance features in the single-cell channels remain subject of debate[17]. Third, missing values can be replaced post hoc by various imputation strategies, which can be highly challenging since peptides may be missing for various reasons, even within individual samples[18–20]. Finally, peptide identity propagation (PIP) provides a powerful approach to transfer peptide sequence information between samples, enabling the assignment of a peptide identifier to a feature even if it had not been selected for fragmentation[8]. PIP can be performed in a feature-based or ion-based manner, both requiring accurate retention time and $m/z$ alignment across samples. Feature-based PIP additionally requires the presence of molecular ions to be observed as isotope peak patterns[21], which can be performed with traditional feature detection algorithms as has been implemented by the match-between-runs (MBR) algorithm in the MaxQuant environment[21,22]. However, the need to recognize isotope peak patterns limits the sensitivity of this approach, thereby preventing feature-based PIP from fully solving the missing-value problem[8]. In addition, false transfers cannot be excluded, requiring specific attention[23]. In contrast, ion-based PIP applies direct ion current extraction (DICE), and only requires the existence of ions within a given retention time and mass-to-charge window, thus enabling sensitive identity propagation[1]. Yet, ion-based PIP is even more dependent on accurate feature alignment to enable narrow DICE windows as otherwise co-eluting species can distort quantity estimations or introduce false positives.

To date, dozens of alignment algorithms for LC–MS data have been proposed of which the majority relies on the fitting of warping models relative to a reference sample[24]. While these approaches allow the correction of systematic deviations between samples, local sample-specific and feature-specific deviations are often overseen. Furthermore, the majority of algorithms simplify the alignment procedure by considering only certain dimensions of the data, e.g. total ion chromatograms (TIC) or extracted ion chromatograms (XIC). Additionally, random mass errors occurring in the mass spectrometer, e.g. caused by changes in electric fields, space-charge effects or temperature are typically ignored[24]. To enable concise DICE windows, the choice of an alignment algorithm that enables systematic as well as local sample-specific and feature-specific corrections in RT but also in $m/z$ dimension is highly important but remains to be established.

Two recent proteomics tools that implement ion-based PIP are DeMix-Q[8] and IonStar[12], both achieving highly reduced missing value rates and improved sensitivity to detect differentially abundant proteins compared to MaxQuant. This shows the potential of DICE, yet a number of fundamental and practical issues remain. For instance, both DeMix-Q and IonStar require large and fixed DICE windows (e.g. defaulting to $m/z$ ± 5 ppm, RT ±1 min) resulting in data deterioration by co-eluting interferences, probably caused by sub-optimal sample alignment. Both DeMix-Q and IonStar use sophisticated alignment algorithms (based on the OpenMS proteomics pipeline[25] and ChromAlign[26] implemented in the commercial tool SIEVE from Thermo Scientific, respectively), however, they only correct for systematic deviations, thereby ignoring potential random local feature-specific deviations. Furthermore, both tools are designed to only process RT and $m/z$ data from Thermo Fisher Scientific Orbitrap mass spectrometers, excluding their use to other vendors and scan modes (e.g. ion mobility (IM) separation). In addition, the running of DeMix-Q and IonStar is not straightforward, requiring the installation of several tools including discontinued commercial applications. This may explain why DICE-based approaches, despite their clear advantages, have not permeated into mainstream proteomics applications.

It is easy to argue why the principle of DICE is advantageous over other PIP methods, allowing sensitive feature detection and making post hoc imputation obsolete, however, its implementation with a comprehensive feature alignment approach, especially in a user-friendly manner, has not been established yet. Therefore, we here present IceR (Ion current extraction Re-quantification), an efficient, robust, and user-friendly label-free proteomics quantification workflow. The method uniquely combines the following features: (1) a hybrid PIP strategy merging feature-based and ion-based PIP, thus combining the strengths of both approaches; (2) robust 2-step feature alignment incorporating global modeling-based and local feature-specific kernel density estimation (KDE)-based alignment, allowing narrow and feature-specific DICE-windows in $m/z$- and RT-space. Thereby this resolves several key limitations associated with other alignment methods[24]; (3) capability to handle tims-TOF data to utilize IM as an additional dimension for feature detection and alignment; (4) sound decoy feature-based scoring schemes to assess the reliability of quantifications and to distinguish the true presence of peptides from random ion occurrences; (5) a superior noise-model-based imputation approach allowing accurate estimations of ratios of low abundance peptides and proteins.

To allow for broad and easy applicability, we have implemented IceR as a user-friendly R-package (https://github.com/mathiaskalxdorf/IceR[27]) and it can be seamlessly integrated with the MaxQuant suite. The software provides a graphical user interface to set up analyses and inspect quality control measures.

We have comprehensively assessed and benchmarked IceR on 4 publicly available and 3 in-house generated data sets including comparisons against DeMix-Q, IonStar, Proteome Discoverer with apQuant, MSFragger with IonQuant, DIA, and IM-enhanced MaxQuant. Furthermore, we have evaluated its performance on published plasma and single-cell proteomics data sets enabling highly increased numbers of reliably quantified proteins, increased data completeness and improved discriminability between single-cell populations allowing de novo reconstruction of a developmental trajectory. Collectively this indicates that IceR is a powerful tool to improve the performance of label-free proteome analysis in a wide variety of applications.

## Results

**Analysis pipeline**. We designed IceR to leverage DICE for peptide quantification and to use it for reliable and sensitive PIP to minimize missing values across proteomic data sets. In addition, since DICE will primarily rescue low-intensity peptides that escaped direct identification by MS2, we aimed to assess the overall gain in sensitivity that could be achieved compared to commonly used approaches for PIP. The IceR approach is schematically summarized in Fig. 1a, automatically proceeding through a number of steps as described in Supplementary Note 1, also illustrated in Supplementary Fig. 1. In brief, IceR starts from lists of peptide features detected in label-free DDA data. These features are aggregated and aligned over all samples (steps 1–3 in Supplementary Fig. 1), and finally, the respective quantities are extracted by DICE from MS raw files. To enable reliable PIP, IceR performs modeling-based RT- and $m/z$-corrections (step 4), similar to MaxQuant, DeMix-Q and IonStar. Next, IceR incorporates a hybrid PIP approach: peptide information is propagated preferably by feature-based PIP and identities are recovered by ion-based PIP only in cases of missing feature detection (step 5). To further improve the reliability of ion-based PIP, IceR performs an additional local alignment step. While step 4 can greatly decrease global variability between samples, local sample-specific and feature-specific heterogeneities might still be missed, which could result in false matches. To account for this, IceR uniquely performs a second sample-specific and feature-specific alignment step based on kernel density estimated ion accumulation maps (steps 6–8). This enables highly robust alignments over samples, and hence allows more narrow DICE windows. Decoy feature-based scoring schemes are applied to assess the reliability of PIPs and quantifications (steps 6, 9–11). Along with a detailed description of all steps of the IceR workflow, Supplementary Note 1 and Supplementary Figs. 2 and 3 illustrate its application to a published spike-in data set (iPRG 2015 study[28]) and assess its performance compared to MaxQuant and DeMix-Q (Supplementary Data 1). Among the most salient findings, (i) IceR could transfer peptide sequence information for nearly twice the number features due to its hybrid PIP approach (supplementary Fig. 2a) and its superior two-step alignment algorithm (Supplementary Fig. 2b, c); (ii) FDR of DICE-based peak selection was estimated at 0.6% (Supplementary Fig. 2d); (iii) on average 85% of all peptide features could be directly quantified by IceR (52% by MaxQuant) (Supplementary Fig. 2a, e); (iv) IceR reduced missing value rates by 12-fold compared to MaxQuant, being at par with DeMix-Q (Supplementary Fig. 2f, g); (v) IceR resulted in most accurate and precise protein abundance ratio estimations (Supplementary Fig. 2h) and enhanced statistical power for DE analyses compared to DeMix-Q (Supplementary Fig. 2i). The results of this initial analysis indicated the highly promising performance of IceR, prompting us to evaluate it in an additional set of use cases.

**Evaluation of IceR based on public spike-in data sets**. Beyond testing against the iPRG2015 study[28] (Supplementary Figs. 2 and 3, supplementary Data 1), we benchmarked IceR against two additional published data sets with increasing complexity.

In the data set published by Ramus et al.[29] (Supplementary Data 2), 48 recombinant human proteins (UPS1 mix) were spiked into yeast lysate at 9 different concentrations ranging from 0.05 to 50 fmol/μL, to test the recovery of a defined set of proteins against a complex background. In total, 883 yeast proteins and 43 UPS1 proteins were identified by MaxQuant (MBR enabled) with at least 2 peptides (Supplementary Fig. 4a). Similar numbers could be achieved when using apQuant or MSFragger. When re-analysing these data with IceR, the fraction of missing values on protein-level could be reduced to 1.8% from 13.0% in MaxQuant, 29.6% in apQuant and 19.3% in MSFragger (Fig. 1b, c). Similarly, IceR could reduce the fraction of missing values on peptide-level compared to the other tools. This enabled almost full quantification of the 45 identified UPS1 proteins in IceR down to the most diluted condition. Coefficients of variation (CV) of quantifications were comparably low between MaxQuant and IceR data (both 4.9% on protein-level, supplementary Fig. 4b, c), despite the fact that more than 2-times more quantification events on peptide-level were available in IceR data (48K vs. 117 K, Supplementary Fig. 4c), the majority of which were of low abundance (Fig. 1b). Interestingly, apQuant and MSFragger showed generally more reproducible peptide quantifications, however, especially in case of apQuant at the cost of fewer quantification events. The receiver operating characteristic (ROC) showed superior performance of IceR over all 36 pairwise DE analyses on protein- and on peptide-level compared to all other tools e.g. when applying a specificity cut-off at 5 % (Fig. 1d, Supplementary Figs. 4f, g, 5). Accordingly, IceR resulted in the highest true positive rates (TPR) as well as cumulative true positives while keeping false positive rates (FPR) and cumulative false positives low (Fig. 1e, f). Imputation of missing values in case of MaxQuant, apQuant and MSFragger data by the bpca-method (Bayesian principal component analysis), which performed best compared to all other tested methods (Supplementary Fig. 4d), true positives could be increased, however, at the cost of increasing false positives as well as inaccurate spike protein abundance ratio estimations (Supplementary Fig. 4e).

In the data set published by Shen et al.[12] (Supplementary Data 3), a complete *E. coli* lysate was spiked into human lysate at 5 ratios (*E. coli*/human), ranging from 3% to 9 %. As this data set was originally used to evaluate the performance of IonStar in recovering low abundance proteins at a proteomic scale, we used it here for its direct comparison to MaxQuant, apQuant, MSFragger, and IceR. Numbers of identified proteins were comparable between all tools, however, IonStar identified >10% fewer proteins (Supplementary Fig. 4h). Missing values on protein-level were between 14.1% and 22.7% in MaxQuant, apQuant, and MSFragger, which was reduced to 0.1% by IceR and 0% in IonStar, thus allowing an (almost) complete quantification of identified *E. coli* proteins by the latter two approaches (Fig. 2a, b). Analysis of quantification variability revealed a comparable CV between MaxQuant, MSFragger, and IceR data (~6% on protein-level) while IonStar and apQuant resulted in more variable quantification (10% and 14% on protein-level) between replicates (Supplementary Fig. 4i, j). Still, IonStar outperformed MaxQuant, apQuant, and MSFragger in detecting differentially abundant spiked proteins in pairwise differential expression analyses as evidenced from the area under the ROC (AUROC) that was increased from 0.73, 0.63, and 0.72, respectively, to 0.95 on protein-level (Fig. 2c). Imputing missing values in MaxQuant,

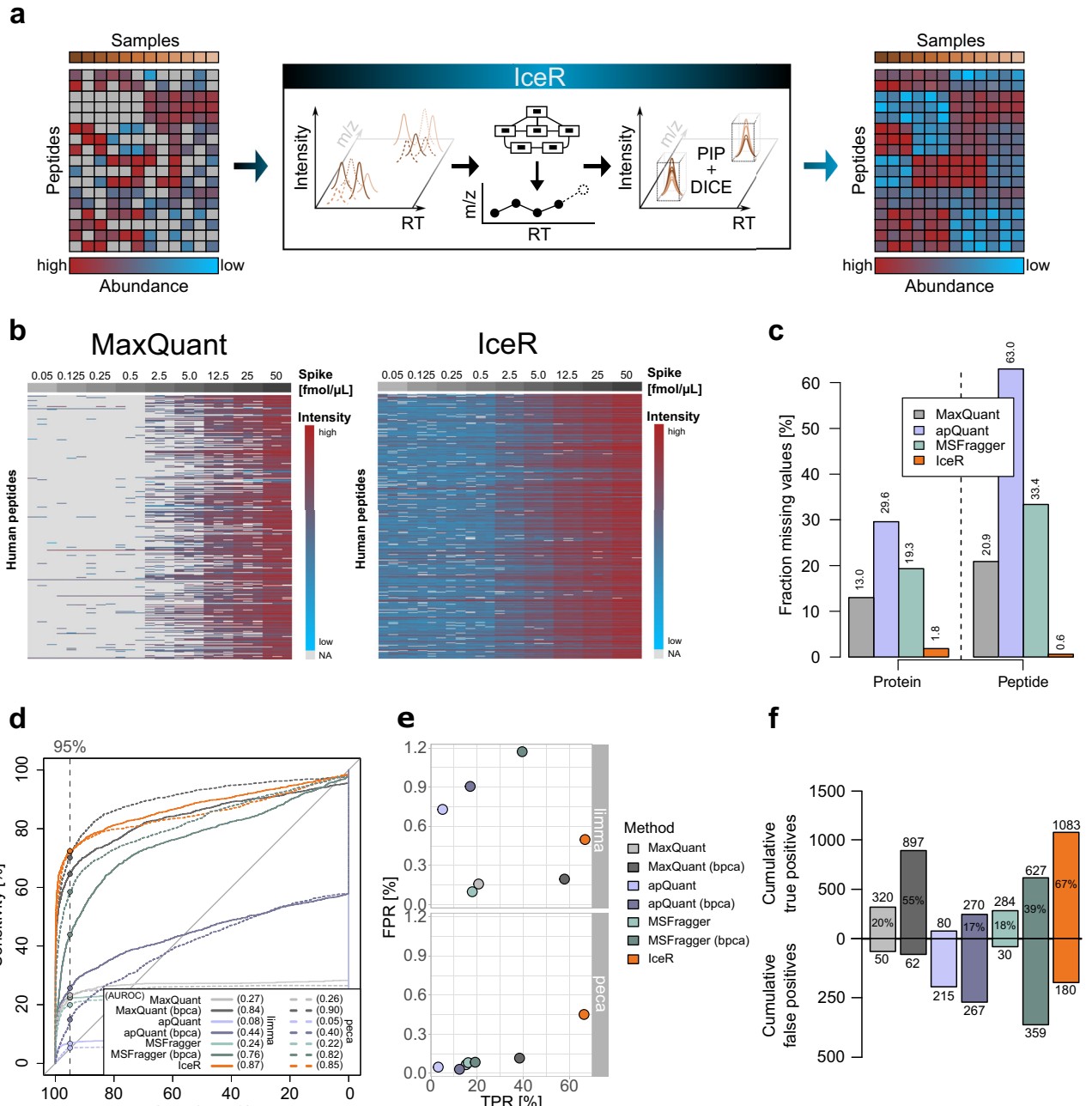

**Fig. 1 IceR enables enhanced sensitivity to detect differentially abundant proteins. a** Label-free DDA proteomics typically results in increasing numbers of missing values with increasing sample sizes due to the stochastic principle of precursor selection for fragmentation. IceR addresses this issue by performing robust peptide feature alignment of samples in *m/z* and chromatographic retention time (RT) space enabling reliable peptide identity propagation (PIP) across samples and highly sensitive and accurate quantification by direct ion current extraction (DICE) quantification. Thereby, IceR highly reduces missing value rates and enables comprehensive, precise and accurate label-free proteomics analyses. **b** Heatmap representation of quantified peptides of 48 spiked proteins at nine spike amounts into the constant background (*n* = 3) in MaxQuant (left) and IceR (right) results. Low abundant peptides are coloured blue, high abundant peptides are coloured red, and missing values are indicated in grey. **c** Fraction of missing values on protein- and peptide-level in MaxQuant (grey), apQuant (purple), MSFragger (green), and IceR (orange) results. **d** Receiver operating characteristics (ROC) over all pairwise differential expression analyses for MaxQuant (light grey), MaxQuant with bpca imputation (dark grey), apQuant (purple), apQuant with bpca imputation (dark purple), MSFragger (green), MSFragger with bpca imputation (dark green), and IceR (orange) on protein-level using limma (solid line) and peptide-level using peca (dashed line). Area under the ROC (AUROC) per condition is indicated. Dashed vertical line and respective dots represent observed sensitivity at 95% specificity per method. **e** True and false positive rates over all (36) pairwise DE analyses using limma or peca for MaxQuant (grey), MaxQuant with imputation (dark grey), apQuant (purple), apQuant with bpca imputation (dark purple), MSFragger (green), MSFragger with bpca imputation (dark green), and IceR (orange). **f** Cumulative true and false positives over all (36) pairwise DE analyses for MaxQuant (grey), MaxQuant with imputation (dark grey), apQuant (purple), apQuant with bpca imputation (dark purple), MSFragger (green), MSFragger with bpca imputation (dark green), and IceR (orange) when using limma. True positive rates are indicated. bpca Bayesian principal component analysis.

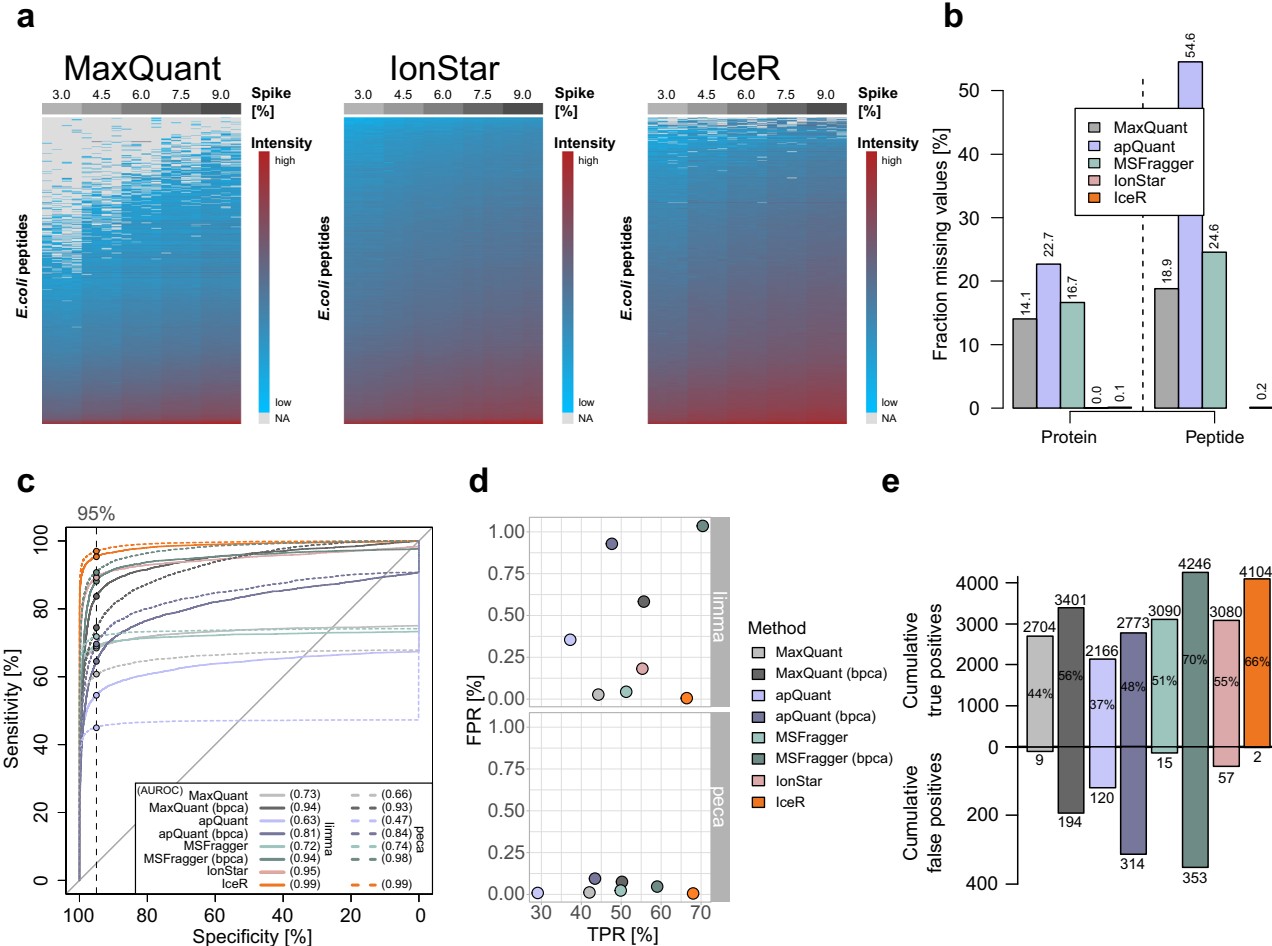

**Fig. 2 IceR outperforms other label-free quantification workflows. a** Heatmap representation of quantified peptides of spiked *E. coli* lysate at five spike amounts into constant background (n = 4) in MaxQuant (left), IonStar (middle) and IceR (right) results. Low abundance peptides are coloured blue, high abundance peptides are coloured red, and missing values are indicated in grey. **b** Fraction of missing values on protein- and peptide-level in MaxQuant (grey), apQuant (purple), MSFragger (green), IonStar (pink), and IceR (orange) results. **c** Receiver operating characteristics (ROC) over all pairwise differential expression analyses for MaxQuant (light grey), MaxQuant with bpca imputation (dark grey), apQuant (purple), apQuant with bpca imputation (dark purple), MSFragger (green), MSFragger with bpca imputation (dark green), IonStar (pink) and IceR (orange) on protein-level using limma (solid line) and peptide-level using peca (dashed line). Area under the ROC (AUROC) per condition is indicated. Dashed vertical line and respective dots represent observed sensitivity at 95 % specificity per method. **d** True and false positive rates over all (10) pairwise DE analyses using limma or peca for MaxQuant (grey), MaxQuant with imputation (dark grey), apQuant (purple), apQuant with bpca imputation (dark purple), MSFragger (green), MSFragger with bpca imputation (dark green), IonStar (pink), and IceR (orange). **e** Cumulative true and false positives over all (10) pairwise DE analyses for MaxQuant (grey), MaxQuant with bpca imputation (dark grey), apQuant (purple), apQuant with bpca imputation (dark purple), MSFragger (green), MSFragger with bpca imputation (dark green), IonStar (pink), and IceR (orange) when using limma. True positive rates are indicated. bpca – Bayesian principal component analysis.

apQuant, and MSFragger data by the bpca-method, which performed best compared to all other tested methods (Supplementary Fig. 4k), reduced but not completely abolished this AUROC difference. In contrast, IceR resulted in an almost perfect sensitivity over all pairwise differential expression analyses, indicated by an AUROC of 0.99 on protein- and peptide-level (Fig. 2c). Accordingly, IceR resulted in highest TPR to FPR and TP to FP ratios over all pairwise DE analyses using limma and peca (Fig. 2d, e, Supplementary Fig. 4m, n, Supplementary Fig. 6). Spiked protein abundance ratios could be accurately estimated by all methods, however, IceR allowed most accurate and precise estimations for high numbers of detected *E. coli* proteins (Supplementary Fig. 4l). In conclusion, IceR clearly outperformed MaxQuant, apQuant, MSFragger, and IonStar by enabling low missing value rates with highly accurate and precise quantifications resulting in the highest TPR at lowest FPR.

**Evaluation of IceR performance at various MS-acquisition conditions.** Experimental parameters in MS are typically varied to maximize the number of identified and quantified proteins, however, this may inflate missing values or deteriorate the reliability of quantification. To evaluate how IceR performs within typical parameter ranges, an *E. coli* lysate was spiked into constant human lysate in six different amounts (0%, 3%, 4.5%, 6%, 7.5% and 9% relative to human, wt/wt, n = 3, supplementary Data 4). Respective samples were analysed by changing one parameter while keeping the other parameters constant. We tested: (1) Top N (Top5, Top10 and Top20); (2) Gradient length (1 h, 1h25 and 2 h); and (3) Sample amount (500, 50, and 10 ng).

In all tested scenarios, IceR outperformed MaxQuant results: It enabled identification of more *E. coli* proteins with at least two quantification events (Fig. 3a, top panel), resulted in much less

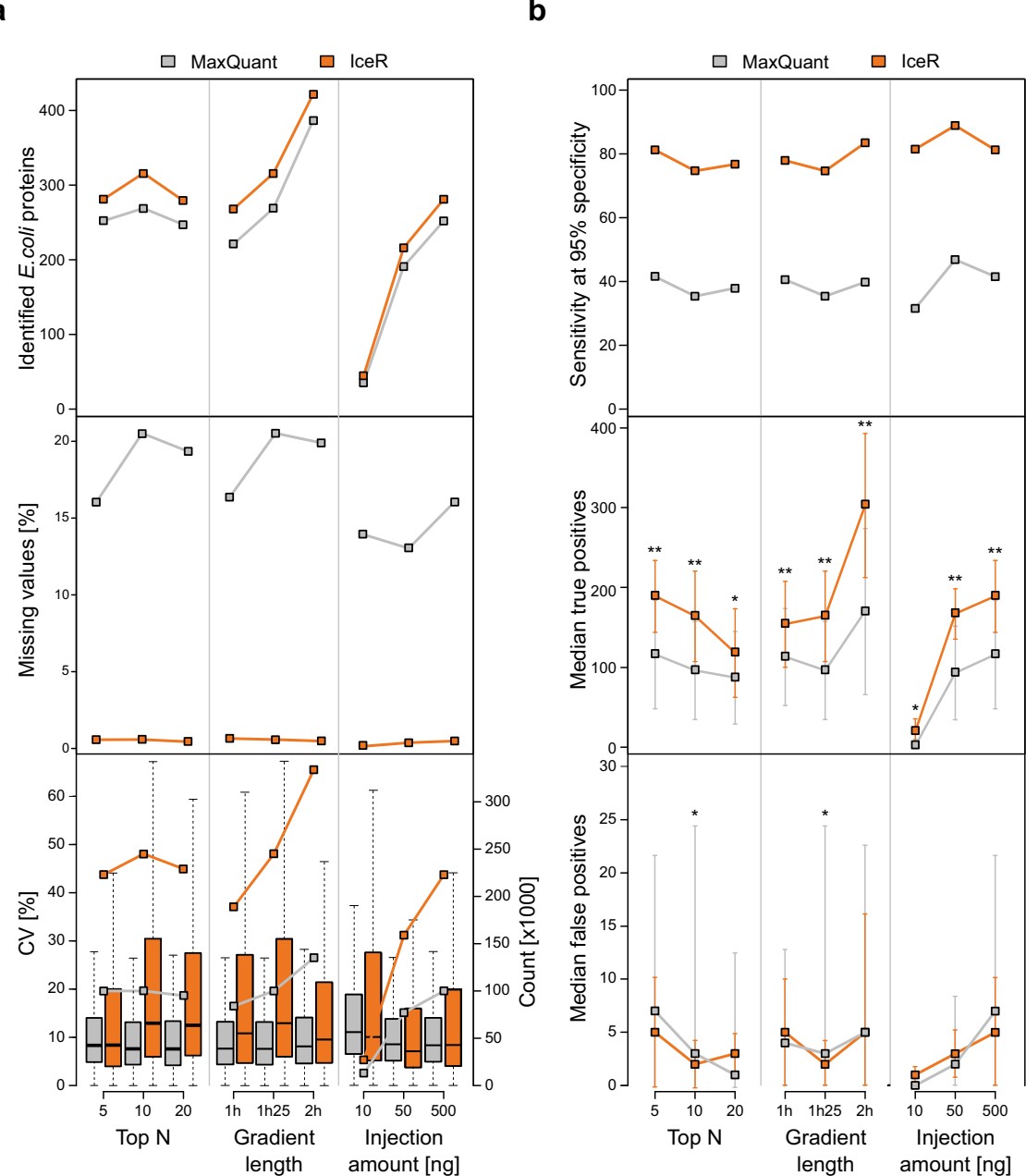

**Fig. 3 IceR enables robustly improved sensitivity for wide ranges of typically varied MS analysis parameters. a** Effect of TopN, gradient length and sample injection amount on numbers of identified *E. coli* proteins with at least two quantification events (top panel), missing values (centre panel) and CVs of quantification (bottom panel) in MaxQuant (grey) and IceR (orange) data. Each condition was analysed in $n = 3$ independent samples. centre line of boxplots, median; box limits, upper and lower quartiles; whiskers, 1.5× interquartile range. **b** As in **a** but showing the effect of parameters on sensitivity (extracted from ROC curves at 95% specificity, upper panel), detection of true positives (center panel) and detection of false positives (bottom panel) over all pairwise differential expression analyses in MaxQuant (grey) and IceR (orange) data. Each condition was analysed in $n = 3$ independent samples. Asterisks indicate significance level which was determined by one-sided $t$-tests based on log-transformed counts: *$p$-value < 0.05, **$p$-value < 0.01.

missing values (Fig. 3a, centre panel) and it more than doubled the number of available quantification data at low CV (Fig. 3a, bottom panel). It doubled sensitivity (Fig. 3b, top panel) and AUROC (Supplementary Fig. 7a, upper row) in DE analyses enabling detection of significantly more true positive spike-in proteins (Fig. 3b, centre panel) while keeping FPR low (Fig. 3b, bottom panel). Spike-in ratios were comparably well estimated by MaxQuant and IceR in all tested scenarios with errors being at median always below 25% (Supplementary Fig. 7d). Generally, increasing the number of MS1 spectra by selecting a lower TopN method or by increasing gradient lengths is especially beneficial

for IceR as it increases the reproducibility of quantifications and sensitivity to detect differentially abundant proteins. IceR further improved sensitivity in case of very low sample injection amounts (Fig. 3b and Supplementary Fig. 4e). Interestingly, median ion counts for quantified peptide features was highest in the 10 ng setup (Supplementary Fig. 7b), and peptide features that were quantified in all three injection amounts revealed that ion counts were only slightly lower in 10 ng than in 500 ng sample injections (Supplementary Fig. 7c). These collective data show that IceR delivers favourable results in a range of commonly used experimental regimes.

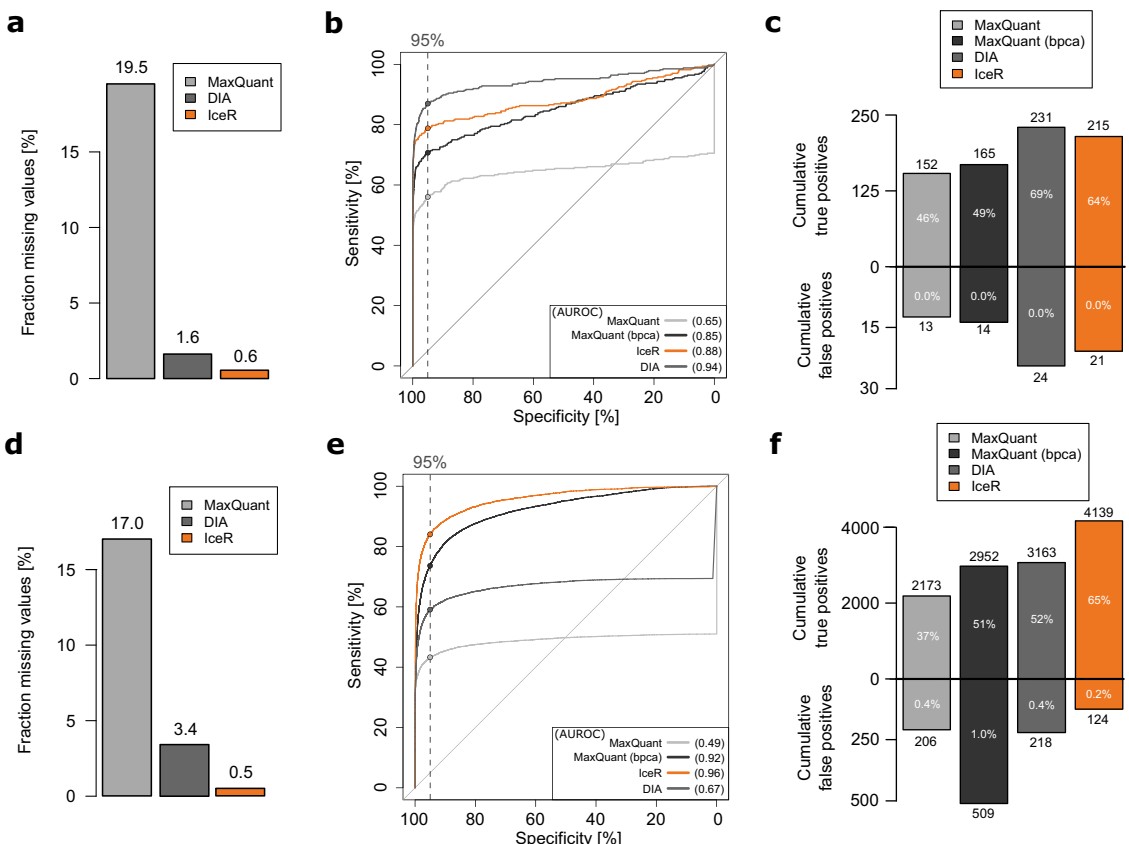

**Fig. 4 IceR enables label-free DDA proteomics with DIA performance. a** Fraction of missing values on peptide-level in MaxQuant (grey), DIA (red) and IceR (orange) results for a publicly available data set of 12 spiked proteins into a constant complex background at 8 concentrations ($n = 3$). **b** Receiver operating characteristic (ROC) over all pairwise differential expression analyses on protein-level in MaxQuant (grey) and on peptide-level in IceR (orange) and DIA (red) data. Corresponding areas under the ROC (AUROC) are indicated. Dashed vertical line and respective dots represent observed sensitivity at 95% specificity per method. **c** Cumulative true and false-positive counts over all (28) pairwise DE analyses for MaxQuant (grey), DIA (dark grey) and IceR (orange). Percentages within bars indicate corresponding true positive and false-positive rates. **d** Fraction of missing values on peptide-level in MaxQuant (grey), DIA (red) and IceR (orange) results for a data set of an *E. coli* lysate spiked into a constant background at 6 concentrations ($n = 3$). **e** Receiver operating characteristic (ROC) over all pairwise differential expression analyses protein-level in MaxQuant (grey) and on peptide-level in IceR (orange) and DIA (red) data. Corresponding areas under the ROC (AUROC) are indicated. Dashed vertical line and respective dots represent observed sensitivity at 95% specificity per method. **f** Cumulative true and false positives over all (15) pairwise DE analyses for MaxQuant (grey), DIA (dark grey) and IceR (orange). Percentages within bars indicate corresponding true positive and false-positive rates.

**Performance of IceR in comparison to DIA label-free proteomics**. Data-independent acquisition (DIA) is an emerging technology for quantitative proteomics and has been shown to be superior in comparison to DDA as it can result in fewer missing values, and lower CVs across replicates[30]. Since IceR performed particularly well with regard to returning low missing value rates (Figs. 1c, 2b and 3a), we here wanted to evaluate the performance of IceR in comparison to DIA. To enable a more comprehensive comparison, we chose two data sets with different complexities: (1) a publicly available data set with 12 non-human proteins spiked into human lysate[31] and (2) an in-house generated data set with a complete *E. coli* lysate spiked into the human lysate.

The first data set generated by Bruderer et al.[31] (Supplementary Data 5) was originally used to introduce a SWATH-MS-type DIA workflow called hyper reaction monitoring (HRM), and to compare its performance against DDA. For that purpose, 12 proteins were spiked into a constant human background introducing protein abundance changes ranging from as little as 10% up to 5000%, collectively resulting in 8 different samples ($n = 3$) and 28 possible pairwise comparisons. All 12 spiked proteins were identified in the DDA and DIA data, and the total numbers of identified proteins and peptides were comparable between MaxQuant, DIA (HRM) and IceR (Supplementary

Fig. 8a, b). The fraction of missing values on peptide-level (19.5% in MaxQuant) could be reduced to 1.6% in DIA and, remarkably, to 0.6% in IceR data (Fig. 4a). Comparison of observed CVs on peptide-level between MaxQuant, DIA and IceR showed the highly reproducible quantification in DIA data. Median CV could be reduced from 16% in MaxQuant data to 8% in DIA data, along with an increase in available quantification events by 58% (Supplementary Fig. 8c). Strikingly, IceR resulted in a 250% increase in available quantification events compared to MaxQuant. Since this included more variable low abundant features, the median CV was slightly higher in IceR results (17.5%). When focusing on confidently quantified peptide features ($p$ value of ion accumulation < 0.05, signal to background ≥ 4), median CV could be reduced to 10%, however, still with ~10% more quantification events compared to MaxQuant data. ROC curves revealed the highly improved performance of IceR (AUROC 0.90) compared to MaxQuant (AUROC 0.66) for detection of differentially abundant proteins (Fig. 4b), approaching the performance of DIA (AUROC 0.96). Of 336 total true positives (100%) over all 28 pairwise DE analyses (adj. $p$ value < 0.01, absolute fold-change > 10%), IceR resulted in 64% cumulative true positives with significant differential abundance, compared to 46% for MaxQuant and 69% for DIA (Fig. 4c).

DIA outperformed IceR especially due to its higher sensitivity for very small abundance differences (<40%, Supplementary Fig. 8d). False-positive rates were generally far below 1% for all methods (Fig. 4c and supplementary Fig. 8e).

Next, we compared the performance of IceR, MaxQuant and DIA in the more complex situation of an *E. coli* lysate spiked into a human lysate at six different amounts as used above (0%, 3%, 4.5%, 6%, 7.5%, and 9%). Samples were analysed in DDA and DIA mode with a 2-h gradient method (Supplementary Data 5). The required spectral libraries for DIA were generated from DDA runs, and hence the total required MS resources for the DIA data set were increased (see the "Methods" section for details). In total, about 400 *E. coli* proteins could be identified based on ~2800 peptides in both data sets (Supplementary Fig. 8f, g). The fraction of missing values was reduced from 17.0% in MaxQuant to 3.4% in DIA and even to 0.5% in IceR (Fig. 4d). Quantifications were most reproducible in DIA results, however, IceR resulted in many more available peptide quantification events (+165%) including more variable low-abundance peptide features (Supplementary Fig. 8h). Filtering IceR data for robust quantifications resulted in CVs comparable to DIA data. Pairwise DE analyses on peptide-level for DIA and IceR revealed the superior performance of IceR (AUROC 0.96) compared to MaxQuant (AUROC 0.49) and even to DIA results (AUROC 0.67) in this data set (Fig. 4e). Similarly, IceR resulted in the highest relative and absolute cumulative true positive rate over all DE analyses, greatly outperforming MaxQuant and DIA (Fig. 4f). The superior performance of IceR is driven by its capability to enable abundance ratio estimations even against low/no spike-in conditions and its improved sensitivity for small abundance ratios compared to MaxQuant (Supplementary Fig. 8i). In contrast, DIA fails to estimate ratios against no spike-in conditions but shows higher sensitivity for low abundance ratios (Supplementary Fig 8i). As before, false-positive rates were generally far below 1% for all methods (Supplementary Fig. 8j).

In summary, IceR enables label-free DDA proteomic analyses with DIA-like performance with regard to sensitivity, absence of missing data, and quantitative accuracy. IceR consistently allows more sensitive detection of differential abundances in comparison to standard DDA data analysis and especially boosts sensitivity in case of comparisons against very low-abundance conditions.

**Addition of IM dimension to label-free proteomics analyses.** Combining IM separation with MS promises improved specificity and accuracy for PIP as recently demonstrated for the MBR algorithm in MaxQuant[32]. Still, tools utilizing the additional IM dimension for ion-based PIP are lacking. Here, we reanalysed our *E. coli* spike-in tool sample set on a timsTOF Pro mass spectrometer with a comparable 2-h gradient method as described above and processed the data using MaxQuant, MSFragger, and IceR (Supplementary Data 4). Proteome Discoverer with apQuant is not designed to process timsTOF Pro data, therefore it was only used for comparison of QE-HF results. However, important to note is that apQuant had to be operated with a relaxed confidence parameter as it otherwise ended up with > 95% missing values. When comparing QE-HF and timsTOF Pro data, the total number of identified proteins could be increased by on average >30% while the total fraction of missing values was decreased from 17% to 11% and 21% to 17% on peptide-level in MaxQuant and MSFragger, respectively (match between runs in both cases enabled) (Fig. 5a). IceR reduced this to <1% missing values for data from both instruments (Fig. 5a). Protein and peptide abundances could be reproducibly estimated by all tools for data from both instruments, however, CVs were generally lower in the case of timsTOF Pro data (Supplementary Fig. 9a, b). ROC curves

based on all (15) pairwise differential expression analyses indicated the superior performance of IceR for QE-HF (Fig. 5b, Supplementary Fig. 9f, g) and timsTOF Pro (Fig. 5c, Supplementary Fig. 9h, i) data as evidenced by highly improved AUROC values of 0.95 and 0.96 for the respective instrument. Imputing missing values in MaxQuant, apQuant, and MSFragger data by the bpca-method, which performed best compared to all other tested methods (Supplementary Fig. 9c), reduced but not completely abolished this AUROC margin of IceR. Furthermore, the improvements came at the cost of highly increased FPR (Fig. 5d). Spiked protein abundance ratios could be accurately estimated by all methods for data from both instruments, however, generally with higher precision in the case of timsTOF Pro data especially when using peca (Supplementary Fig. 9d, e). On an absolute scale, IceR enabled the highest cumulative true-positive counts at the lowest cumulative false-positive counts across all pairwise DE analyses for data from both instruments (Fig. 5e). Interestingly, IceR even enabled the detection of more true positives in QE-HF data compared to MaxQuant and MSFragger in timsTOF Pro data despite the overall higher protein coverage by the latter instrument.

In summary, IceR highly benefits from the additional IM dimension, lifting its sensitivity far above the workflows of MaxQuant and MSFragger. Furthermore, here we showed the use of ion-based PIP on timsTOF Pro data, which as yet was not available in any other quantitative workflow.

**Application of IceR to plasma proteomics data.** Blood plasma is an attractive and easily accessible body fluid for proteomics-based biomarker discovery, however, data completeness across samples tends to be low, due in part to the high dynamic range of protein abundance. Here, we applied IceR to a publicly available plasma proteome data set of 32 finger prick samples acquired from one person over 8 consecutive days in short (20-min) LC–MS analyses[33] (Supplementary Data 6). Originally, 257 proteins were quantified with at least two peptides by MaxQuant (MBR enabled) with a missing value rate of 11% of the proteins per sample[33]. In comparison, IceR enabled the identification of 279 proteins where the fraction of missing values per sample could be reduced to at median 2%. The highly improved data completeness from IceR enabled full quantification of 248 proteins (89% of all identified proteins in IceR data) over all 32 samples while Max-Quant only allowed full quantification of 195 proteins (79% of all identified proteins in MaxQuant data, Fig. 6a). The 53 additional proteins now rescued by IceR with 100% completeness showed reproducible quantification levels over all samples and ranged from low to high overall abundance (Fig. 6b) indicating that the IceR workflow goes far beyond simple missing-value imputation. These proteins included several important blood biomarkers like the indicators for myocardial infarction LDHA/LDHB, the coagulation factor F7, the liver injury marker CPS1, the infection marker CRP, the metabolic syndrome marker FABP5[34] and the inflammatory predictor GSN[35].

In summary, the improved sensitivity of IceR allows robust and reproducible quantification of more proteins with better data completeness in plasma proteome profiling data, thus displaying a more comprehensive portrait of a person's health state.

**Application of IceR to label-free single-cell proteomics data.** As demonstrated above, IceR allows increased sensitivity and data completeness even at very low sample injection amounts (Fig. 3 and Supplementary Fig. 7). Since these are critical properties for low-input applications, we reasoned that IceR could be suitable to enhance sampling depth and data completeness of label-free single-cell proteomics data. To evaluate this, we selected a

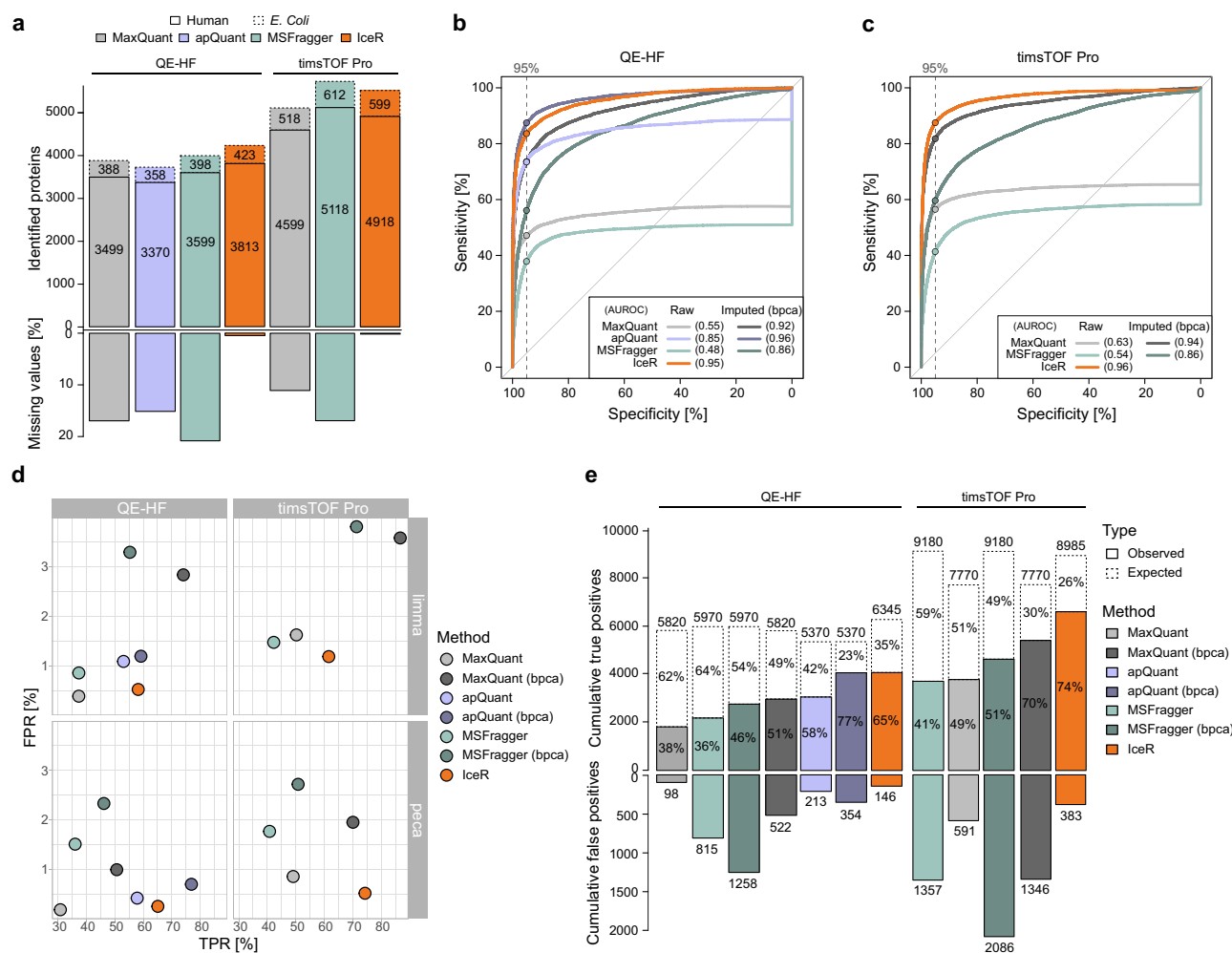

**Fig. 5 Application of IceR to timsTOF Pro proteomics data. a** Comparison of numbers of identified human (solid box) and *E. coli* (dashed box) proteins as well as fraction of missing values in the in-house generated *E. coli* spike-in data set analysed on a Q Exactive HF (QE-HF) and a timsTOF Pro followed by peptide and protein quantification using MaxQuant (grey), apQuant (purple), MSFragger (green) or IceR (orange). **b** Receiver operating characteristics (ROC) over all (15) pairwise differential expression analyses on QE-HF data analysed by MaxQuant (light grey), MaxQuant with bpca imputation (dark grey), apQuant (purple), apQuant with bpca imputation (dark purple), MSFragger (green), MSFragger with bpca imputation (dark green), or IceR (orange) on peptide-level using peca. Area under the ROC (AUROC) per condition is indicated. Dashed vertical lines and respective dots represent observed sensitivity at 95% specificity per method. **c** As in **b** but showing results for timsTOF Pro data. **d** True and false-positive rates over all (15) pairwise DE analyses using limma or peca for QE-HF and timsTOF Pro data for MaxQuant (grey), MaxQuant with imputation (dark grey), apQuant (purple), apQuant with bpca imputation (dark purple), MSFragger (green), MSFragger with bpca imputation (dark green), and IceR (orange). **e** Cumulative true and false positives over all (15) pairwise DE analyses for QE-HF and timsTOF Pro data for MaxQuant (grey), MaxQuant with bpca imputation (dark grey), apQuant (purple), apQuant with bpca imputation (dark purple), MSFragger (green), MSFragger with bpca imputation (dark green), and IceR (orange) when using peca. True positive rates are indicated. bpca—Bayesian principal component analysis.

published data set from Zhu et al. [36] (Supplementary Data 7) where single developing hair and progenitor cells were isolated from utricles of embryonic chickens. Cells were separated by fluorescence-activated cell sorting (FACS) into FM1-43$_{high}$ hair cells and FM1-43$_{low}$ progenitor cells and subsequently processed using the nanoPOTS[37] approach. In total, 10 single FM1-43$_{high}$ hair cells and 10 single FM1-43$_{low}$ progenitor cells were analysed in one batch. As two single hair and four single progenitor cell samples were previously determined to be empty[36], these were here excluded from subsequent analyses. Along with the single-cell samples, pools of 20 FM1-43$_{high}$ cells ($n = 3$) and pools of 20 FM1-43$_{low}$ cells ($n = 3$) were used as matching libraries to boost identification rates in single-cell analyses. Nevertheless, and in line with the published data[36], on average only 23 proteins (min: 5, max: 53) and 103 peptides (min: 35, max: 231) could be quantified per single cell by MaxQuant (Fig. 7a). In contrast,

when applying IceR to these data, on average 433 proteins (min: 417, max: 454) and 1602 peptides (min: 1587, max: 1615, mean features: 2665) could be quantified per single-cell sample (Fig. 7b and Supplementary Fig. 11a). The critical contributor to this improvement is IceR's distinct capability to rescue low-abundance features, thus decreasing the rate of missing values from 95.1% to 8.6% (protein-level) and from 95.3% to 3.6% (peptide-level), boosting both the number of identified/quantified peptides/proteins and data completeness (Fig. 7c). Reproducibility of quantification was comparable between MaxQuant and IceR results, however, IceR enabled in excess of 14-fold and 18-fold more quantification events on protein and peptide-level, respectively, compared to MaxQuant outputs (Supplementary Fig. 11b, c). t-distributed stochastic neighbour embedding (tSNE[38]) revealed clearer separation of hair and progenitor cells in IceR results (Silhouette score of 0.73) compared to MaxQuant

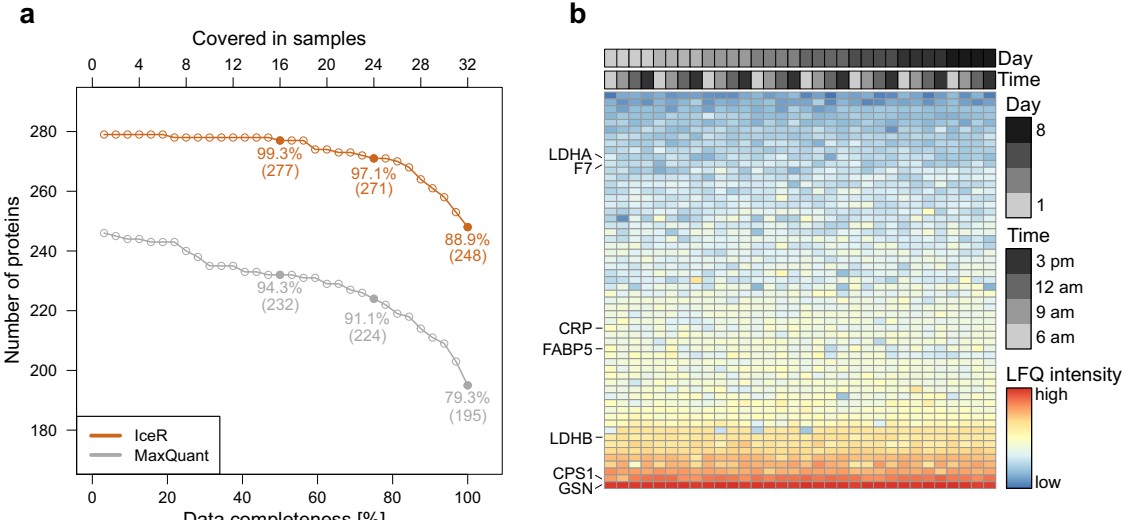

**Fig. 6 Application of IceR to a plasma proteomics data set. a** Numbers of fully quantified proteins with an increasing number of samples for MaxQuant (grey) and IceR (orange) in plasma proteomics analyses of 32 finger prick samples acquired from one person over 8 consecutive days. Relative fractions of fully quantified proteins at 50%, 75% and 100% data completeness are highlighted. **b** LFQ intensities of proteins without missing values in IceR but incomplete quantification in MaxQuant. Samples are ordered per sample acquisition day and time. Colours indicate protein abundance estimations. Seven important plasma protein markers are highlighted.

outputs (Silhouette score of 0.53, Fig. 7d, e) demonstrating the improved data quality after requantification by IceR. This was even more obvious when performing DE analysis between single hair cells and single progenitor cells. While in MaxQuant data differential abundance could be tested only for 37 proteins due to the high data sparsity (Supplementary Fig. 11d), IceR enabled differential abundance testing for 491 proteins and could detect increased expression of proteins known to be enriched in hair cells, including CALB2, CKB, and CRABP1 (Fig. 7f). Unsupervised de novo chronological ordering of cells using the R package CellTrails[39] segregated all cells according to their FM1-43 uptake (Fig. 7g and Supplementary Fig. 11e). Further, developmental trajectories in MaxQuant and IceR results were used to examine protein expression dynamics as a function of developmental pseudotime (Fig. 7h and Supplementary Fig. 11f). Crucially, the highly improved data completeness after requantification by IceR allowed analysis of protein expression dynamics for 6-fold more proteins (413 vs. 66), including proteins known to be enriched in progenitor cells (e.g. TPM3, AGR3, and TMSB4X) and in hair cells like (e.g. OTOF, CALB2, MYO6, CKB, AK1, CRABP1, and GAPDH) (Fig. 7i). Accumulated ion intensities in IceR-selected DICE windows followed the expected developmental trajectory, as observed for peptides originating from OTOF and TPM3 (Fig. 7j) and several other proteins (Supplementary Fig. 11g, h), and clearly distinguished progenitor from hair cells. Enhanced sensitivity of protein detection enabled by IceR now allowed detection of additional proteins with systematic expression changes along the developmental pseudotime axis agreeing with previously published RNA expression[36]. These overlaps included ARF4, SYN3, STARD10, ATP6V1E1, PGM2L1 and RDX (Fig. 7h). The latter is described to anchor cytoskeletal actin of stereocilia to hair cell membranes[40]. Interestingly, RDX showed the highest transcript and protein expression midway through the developmental trajectory (Fig. 7i), suggesting that a maturation-specific transient expression of this protein may be required for proper functionality. Importantly, with the exception of SYN3, none of the above-mentioned proteins could be revealed in the original paper either with regard to a differential or pseudotime-dependent protein expression. This convincingly demonstrates that re-quantification by IceR uncovers biology that

has remained hidden after analysis by conventional data analysis tools. In addition, it shows that DICE-based analysis via IceR is a promising strategy to boost the performance of single-cell proteomics in general.

## Discussion

Our results show that IceR can greatly increase data completeness and quality in DDA-based quantitative label-free proteomics. Its greatly enhanced sensitivity and specificity could be demonstrated for a broad range of published and newly generated data sets demonstrating its universal applicability. IceR rests on two key characteristics: (1) it combines feature-based and ion-based PIP, leveraging the strengths of both principles for robust and sensitive PIP; (2) it introduces a feature alignment approach that includes a correction for both systematic and local deviations such as sample-specific and feature-specific random variation in mass, RT and (unique for IceR) IM. Importantly, alignment parameters are automatically estimated from the data across all samples, instead of requiring the user to specify a reference sample which introduces yet another dependence of results on a user-defined parameter. Finally, consistent quantification of features across all samples is achieved by DICE using the same feature-specific window sizes instead of depending on signal intensities that are deformed by the used warping function. Many of these aspects are not taken into account in most commonly used alignment procedures[24], however, they critically contribute to the performance of IceR. In contrast to previous DICE-based tools, IceR is offered as a comprehensive yet user-friendly R-package with an intuitive graphical user interface (Supplementary Fig. 12) that requires minimal input from the user. To illustrate this, it is worth mentioning that default settings were used for the analysis of all described data sets, showing that robust performance of IceR can be achieved across a wide range of instruments (various orbitraps and timsTOF-Pro), LC gradients (20–180 min) and sample types (plasma, bulk cell lysates, single cells).

Because of these reasons, we expect that IceR can be easily adopted and that it will be of great value for the proteomics community fitting in firmly established and broadly-used DDA-based proteomics workflows. Furthermore, IceR improved the

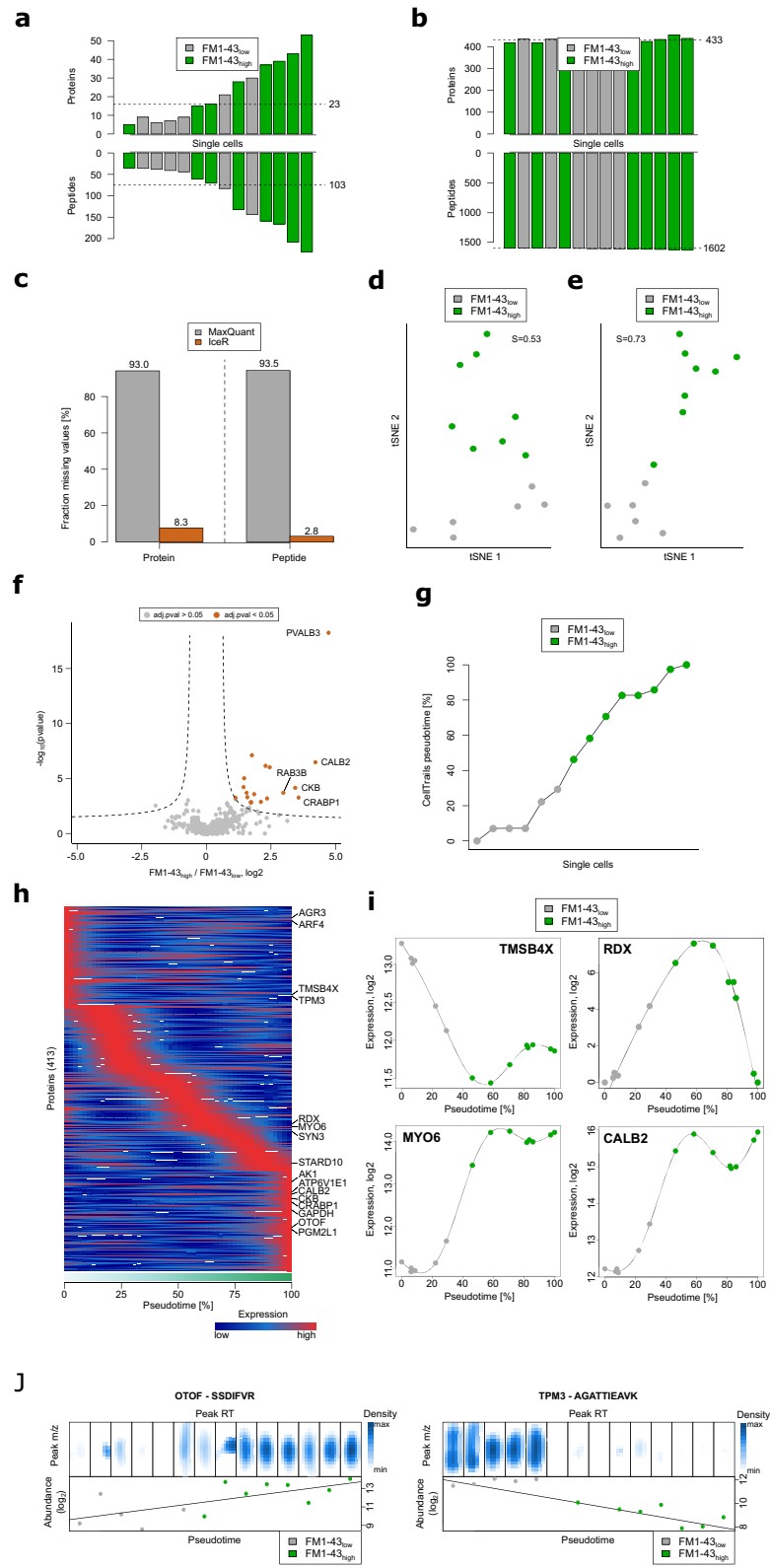

results obtained from a timsTOF Pro instrument, indicating that DICE-based analysis with IceR should be a valuable component in the emerging application of IM mass spectrometry in proteomics.

We envision that IceR can have a strong impact in two main directions, that both receive considerable interest in present-day proteomics, namely biomarker discovery and low-input analysis.

By necessity, biomarker discovery requires large sample cohorts, where, problematically, DDA-based proteomics returns decreasing numbers of fully quantified proteins with increasing cohort size[12]. We have shown here that IceR improved this situation in a series of plasma samples by simultaneously increasing proteome sampling depth and data completeness. Importantly, the property that IceR reduces missing value rates to those seen in DIA, and

**Fig. 7 IceR boosts the completeness and quality of single-cell proteomics data. a** Numbers of proteins (upper panel) and peptides (lower panel) and their average (dashed line) quantified by MaxQuant per single hair (FM1-43$_{high}$, green) and progenitor (FM1-43$_{low}$, grey) cell sample. **b** As in **a**, but after data reprocessing by IceR. **c** Fraction of missing values on protein-level in peptide-level in MaxQuant (grey) and IceR (orange) results. **d** Dimensional reduction of protein abundance estimations in MaxQuant data by t-distributed stochastic neighbour embedding (tSNE). Progenitor cells (FM1-43$_{low}$) are coloured in grey. Hair cells (FM1-43$_{high}$) are coloured in green. Silhouette score (S) is indicated. **e** As in **d** but after data reprocessing by IceR. **f** Volcano plot showing detected significantly (orange, adj. *p* value < 0.05, absolute fold-change > 2) differently abundant proteins between single hair cells and single progenitor cells in IceR data. Significance cut-off is indicated by a dashed line. **g** Chronological ordering of single-cells as a function of CellTrails' inferred pseudotime from IceR data. Single-cells are coloured according to their FM1-43 uptake. **h** Scaled expression dynamics over pseudotime for all analysed proteins in IceR data based on generalized additive models (GAM). Low and high temporal protein expression is indicated by blue and red colour tones, respectively. **i** Absolute expression dynamics of log2 expression levels as a function of pseudotime for various proteins. Single-cells are coloured according to their FM1-43 uptake. **j** Ion density in IceR-selected DICE windows per single-cell sample (upper panels) for the peptides VLTLDLYK and AGATTIEAVK of OTOF (hair cell marker protein) and TPM3 (progenitor cell marker protein), respectively. Corresponding peptide abundances (lower panels) are ordered by CellTrails' inferred pseudotime.

that it primarily rescues low-abundance proteins, demonstrates its potential to open up reservoirs of clinically relevant proteins in DDA data. In the area of low-input proteomics, the field enjoys considerable excitement with early examples demonstrating proof-of-principle of single-cell proteome analysis[16,37,41]. This is the combined result of miniaturized sample preparation, narrow-bore chromatography, and increased sensitivity in mass spectrometry, in conjunction with novel workflows such as TMT multiplexing with booster channels[42–44]. The fact that improved bioinformatics solutions have a major role to play in innovative single-cell strategies has remained underexposed, yet our data showed that re-analysis of an existing single-cell data set by IceR boosted the number of consistently quantified proteins from a few dozen to several hundreds. Indeed, this represents an improvement that has been difficult to achieve by any experimental innovation. As an upshot, highly increased sensitivity allowed the detection of many more differentially expressed proteins and allowed analyses of protein expression dynamics in greater detail. Importantly, this was done directly in single cells, thus indicating that label-free proteomic approaches can be a valuable alternative to TMT multiplexing approaches in the context of single-cell proteomics, avoiding shortcomings and controversies associated with the presence and magnitude of booster channels[17]. In conclusion, IceR contributes to improved analysis of proteomic data in many experimental settings ranging from mainstream proteome characterization to biomarker discovery and low-input proteomics, where IceR could lift single-cell proteomics from infancy to early childhood.

## Methods

**Generation of an in-house tool spike-in data set**. HeLa cells were pelleted from cell culture, snap-frozen in liquid nitrogen and stored at −80 °C. Cell pellet was reconstituted in 100 µL of 0.1% RapiGest SF Surfactant (Waters) in 100 mM triethylammonium bicarbonate (TEAB, Sigma-Aldrich) and 1× protease inhibitor cocktail (PIC, cOmplete, Sigma-Aldrich). Each sample was probe-sonicated for 4 × 15 s at 10% frequency with storage on-ice between cycles of homogenization (Branson Digital Sonifier). Samples were centrifuged at 15,000 × *g*, 4 °C for 30 min to pellet any remaining cell- and tissue-debris, followed by transfer of the supernatant into new reaction tubes and protein quantification using a bicinchoninic acid assay (BCA, Pierce–Thermo Scientific). Proteins were denatured for 5 min at 95 °C. Disulfide-bonds were reduced with dithiothreitol (DTT, 5 mM final concentration, Biomol) at 60 °C for 30 min. Cysteine residues were alkylated using chloroacetamide (CAA, 15 mM final concentration, Sigma-Aldrich) at 23 °C for 30 min. Reduced and alkylated proteins were digested overnight at 37 °C in a table-top thermomixer at 500 rpm using sequencing-grade modified trypsin (Promega) in ddH$_2$O. Upon overnight protein digestion, each sample was acidified to a final concentration of 1% trifluoroacetic acid (TFA, Biosolve Chimie) and incubated at 37 °C and 500 rpm for 30 min, in order to cleave and precipitate RapiGest. Subsequently, samples were centrifuged at 15,000 × *g*, at 23 °C for 30 min to pellet the RapiGest precipitate and recover the peptide-containing supernatant to a new reaction tube. MS injection-ready samples were stored at −20 °C. *E. coli* lyophilized sample (Bio-Rad) was re-suspended in ddH$_2$O to achieve a stock concentration of 2 µg/µL. 100 µL (200 µg) were incubated at 95 °C for 5 min, followed by reduction and alkylation using DTT (10 mM final concentration) at 37 °C for 1-h and CAA

(40 mM final concentration) at 23 °C for 45 min at 500 rpm. Reduced and alkylated proteins were digested overnight at 37 °C in a table-top thermomixer at 700 rpm using sequencing-grade modified trypsin (Promega) in ddH$_2$O. Upon overnight protein digestion, each sample was acidified to a final concentration of 1% TFA (Biosolve Chimie). MS injection-ready samples were stored at −20 °C. Spike-in samples were prepared by mixing HeLa sample with 0%, 3%, 4.5%, 6%, 7.5% or 9% (wt/wt) of *E. coli* sample (*n* = 3).

For the QE-HF acquisition, peptides were separated using the Easy NanoLC 1200 fitted with a trapping (Acclaim PepMap C18, 5 µm, 100 Å, 100 µm × 2 cm, Thermo Fisher Scientific) and an analytical column (nanoEase MZ BEH C18, 1.7 µm, 130 Å, 75 µm × 25 cm, Waters). The outlet of the analytical column was coupled directly to a Q-Exactive HF Orbitrap (Thermo Fisher Scientific) mass spectrometer. Solvent A was ddH$_2$O (Biosolve Chimie), 0.1% (v/v) FA (Biosolve Chimie) and solvent B was 80% acetonitrile (ACN, Pierce–Thermo Scientific) in ddH$_2$O, 0.1% (v/v) FA. The samples were loaded with a constant flow of solvent A at a maximum pressure of 800 bar, onto the trapping column. Peptides were eluted via the analytical column at a constant flow of 0.3 µL/min at 55 °C using three different methods as described below. *2-h method*: During the elution, the percentage of solvent B was increased in a linear fashion from 3% to 8% in 4 min, then from 8% to 10% in 2 min, then from 10% to 32% in a further 68 min, and then to 50% B in 12 min. Finally, the gradient was finished with 8 min at 100% solvent B, followed by 11 min 97% solvent A. *1-h 25 min method*: During elution, the percentage of solvent B was increased linearly from 4% to 5% in 1 min, then from 5% to 27% in 30 min, and then from 27% to 44% in a further 5 min. Finally, the gradient was finished with 10.1 min at 95% solvent B, followed by 13.5 min at 96% solvent A. *1-h 10 min method*: During elution, the percentage of solvent B was increased linearly from 3% to 8% in 4 min, then from 8% to 10% in 2 min, and then from 10% to 32% in a further 17 min, and then to 50% B in 3 min. Finally, the gradient was finished with 8 min at 100% solvent B, followed by 11 min at 97% solvent A. Peptides were introduced into the mass spectrometer via a Pico-Tip Emitter 360 µm OD × 20 µm ID; 10 µm tip (New Objective) and at a spray voltage of 2 kV. The capillary temperature was set at 275 °C. Full scan MS spectra with mass range *m/z* 350–1500 were acquired in the Orbitrap with a resolution of 60,000 FWHM. The filling time was set to a maximum of 50 ms with automatic gain control (AGC) target of 3 × 10$^6$ ions. The top 5, top 10, or top 20 most abundant ions per full scan were selected for an MS$^2$ acquisition. Isotopes, unassigned charges, and charges of +1 or >+8 were excluded. The dynamic exclusion list was with a maximum retention period of 15 s (1 h 10 min) or 25 s and a mass tolerance of plus and minus 10 ppm. For MS$^2$ scans, the resolution was set to 15,000 FWHM with AGC of 1 × 10$^5$ ions and a maximum fill time of 50 ms. The isolation window was set to 2 Th, with a fixed first mass of *m/z* 110, and a stepped collision energy of 26.

For the timsTOF Pro PASEF acquisition of *E. coli* spike-in samples, peptides were separated using the Bruker nanoElute system fitted with an analytical column (Aurora Series Emitter Column with CSI fitting, C18, 1.6 µm, 75 µm × 25 cm) (Ion Optics). The outlet of the analytical column with a captive spray fitting was directly coupled to a timsTOF Pro (Bruker) mass spectrometer using a captive spray source. Solvent A was ddH$_2$O (Biosolve Chimie), 0.1% (v/v) FA (Biosolve Chimie), 2% ACN (Pierce, Thermo Scientific), and solvent B was 100% ACN in ddH$_2$O, 0.1% (v/v) FA. The samples were loaded at a constant maximum pressure of 900 bar. Peptides were eluted via the analytical column at a constant flow of 0.4 µL per minute at 55 °C. During the elution, the percentage of solvent B was increased in a linear fashion from 2% to 17% in 60 min, then from 17% to 25% in 30 min, then from 35% to 37% in a further 10 min, and then to 95% in 10 min. Finally, the gradient was finished with 10 min at 95% solvent B. Peptides were introduced into the mass spectrometer via the standard Bruker captive spray source at default settings. The glass capillary was operated at 3500 V with 500 V endplate offset and 3 L/min dry gas at 180 °C. Full scan MS spectra with mass range *m/z* 100 to 1700 and a 1/*k*0 range from 0.6 to 1.6 V s/cm$^2$ with 100 ms ramp time were acquired with a rolling average switched on (10×). The duty cycle was locked at 100% and the TIMS mode was enabled. All timsTOF and nanoElute methods were default provided by Bruker.

For the DIA acquisition of *E. coli* spike-in samples, gradient and method settings were the same as described for the DDA runs unless otherwise stated. Full scan MS spectra with mass range *m/z* 400 to 1200 were acquired in the Orbitrap with a resolution of 60,000 FWHM. The filling time was set to a maximum of 20 ms with AGC target of $3 \times 10^6$ ions. For the DIA scans, a resolution was set to 30,000 FWHM, with AGC target of $3 \times 10^6$ ions, a fixed first mass of 200 *m/z*, stepped collision energy of 27, and a loop count of 34 with an isolation window of 24.3 *m/z*.

**Data repositories**. Raw LC–MS/MS data of the individual publicly available data sets were downloaded from respective sources:

iPRG 2015 dataset:[28] MSV000079843 [https://massive.ucsd.edu/ProteoSAFe/dataset.jsp?task=eccf4bd3e86a4f79af468b0010eb80b0] or ftp://iprg_study:ABRF329@ftp.peptideatlas.org/
Ramus et al. dataset:[29] PXD001819.
Shen et al. dataset:[12] PXD003881.
Bruderer et al. dataset:[31] PASS00589.
Zhu et al. dataset:[36] PXD014256.
Geyer et al. dataset:[33] PXD002854.

The in-house generated LC–MS/MS raw files have been deposited to the ProteomeXchange Consortium via the PRIDE partner repository[45] with the dataset identifier PXD019777 and PXD021425.

**Data preprocessing**. Raw files were processed using MaxQuant (version 1.5.1.2 or 1.6.14.0), apQuant 3.1 with Proteome Discoverer 2.4, MSFragger 3.1.1 with FragPipe 14.0 or Spectronaut 14.0. The searches for the individual data sets were performed against the following databases concatenated with reversed sequences:

iPRG 2015 dataset:[28] supplied iPRG2015 database (6634 entries).
Ramus et al. dataset:[29] UniProt database consisting of reviewed *S. cerevisiae* and human UPS1 proteins (August 2019, 9805 entries).
Shen et al. dataset:[12]—UniProt database consisting of reviewed human and *E. coli* proteins (September 2019, 24,644 entries).
Bruderer et al. dataset:[31] UniProt database consisting of reviewed human proteins (August 2017, 20,214 entries) including the 12 spiked proteins.
Zhu et al. dataset:[36] UniProt database consisting of chicken proteins (April 2020, 34,726 entries).
Geyer et al. dataset:[33] UniProt database consisting of reviewed human proteins (March 2020, 20,365 entries).
In-house generated *E. coli* spike-in data set: UniProt database consisting of reviewed human and *E. coli* proteins (September 2019, 24,644 entries).

The search engines of the respective tools were used with the following search criteria: enzyme was set to trypsin/P with up to 2 missed cleavages. Carbamidomethylation (C) was selected as a fixed modification; oxidation (M), acetylation (protein N-term) were set as variable modification. Match between runs was enabled for all DDA analysis tools. All tools were used with default parameters with the exception for apQuant for the in-house generated QE-HF spike-in data set where we had to set apQuant Confidence Medium Cutoff to 0.2 as otherwise almost all peptide quantifications were filtered. In the case of MaxQuant, quantification intensities were calculated by the default fast MaxLFQ algorithm with a minimal ratio count set to 1 or 2. Require MS/MS for LFQ comparisons was disabled. Peptide and protein hits were filtered at a false discovery rate (FDR) of 1%, with a minimal peptide length of 7 amino acids. Second peptide search for the identification of chimeric MS2 spectra was enabled. Not mentioned MaxQuant settings were left as default.

**IceR workflow**. IceR requires raw files to be converted into centroided mzXML files. This can either be done by the user beforehand or IceR triggers the conversion using the ProteoWizard tool msConvert if installed. The complete workflow of IceR is illustrated in Supplementary Fig. 1 and includes 13 steps that are implemented in an R-package:

1. Features detected by MaxQuant are aligned over samples. For that purpose, deviations of observed retention time (RT) and calibrated *m/z* for peptides identified in samples are determined. The RT feature alignment window is defined as $1.5 \times$ inter-quartile range of observed absolute RT deviations of peptides between samples. The *m/z* feature alignment window is defined as the inter-quartile range of observed absolute *m/z* deviations of peptides between samples. If a minimal RT-window or *m/z*-window is defined by the user and the observed window is smaller than the specified window, the user-defined minimal window is used. If the final RT-window and/or *m/z*-window is already specified by the user, the user-defined parameter is used.
2. Features detected by MaxQuant (MaxQ features from allpeptides.txt) are aggregated using the determined RT- and *m/z*-alignment windows. First, MaxQ features with the same peptide sequence, same charge-state and same PTM over samples are aggregated and a new IceR feature with median RT and *m/z* of these MaxQ features are defined. Peptide features from samples with *m/z* or RT deviating from these medians by more than the defined alignment windows are excluded. Detected unknown MaxQ features within

the alignment windows but without sequencing information are added. RT peak widths per IceR feature are defined as the maximum RT peak width observed for any of the aggregated MaxQ features. Overlapping IceR features (default: delta mass <0.002 Da) are merged. Optional: Unknown features (without sequence information) which are left after peptide feature aggregation can be aggregated accordingly.
3. For every IceR feature, a decoy feature is generated. Here, alignment windows multiplied by 5 are added to the IceR feature RT and *m/z*. By default: For every IceR feature an expected +1-isotope feature is added. Here it is assumed that the isotope features should show an *m/z* shift of roughly 1.002 Da per charge.
4. For every IceR feature and every sample, the individual *m/z* correction factor is extracted. For samples without an observed MaxQ feature in the respective IceR feature, the *m/z* correction factor has to be estimated. For that purpose, random forest models (RFs, R-package randomForest, version 4.6.14) are trained per sample based on 80% of available MaxQ features with RT, *m/z*, charge and resolution as predictors and deviation of uncalibrated to calibrated *m/z* as response factors. Number of trees is set to 100. Number of variables randomly sampled as candidates at each split is set to 4. The minimal size of terminal nodes is set to 100. Trained models are validated using the remaining 20% of available data. Next, for every IceR feature and every sample, the individual RT deviation of observed MaxQ features from IceR RT (median) are extracted. For samples without an observed MaxQ feature in the respective IceR feature the RT deviation has to be estimated. For that purpose, RT-dependent generalized additive models (GAMs, R-package mgcv, version 1.8.31) are fitted per sample to deviations between IceR feature RT and observed MaxQ feature RTs.
5. Peptide sequence information within IceR features is propagated (feature-based PIP) from sequenced MaxQ features between samples.
6. Background noise, which is expected per IceR feature quantification, is estimated by counting and summing up intensities of ions that are falling into decoy feature DICE windows. These windows are defined as *m/z* of IceR feature ±*m/z* alignment window and RT ± RT peak width/2. Finally, RT-dependent GAMs are fitted to the observed decoy feature intensities and decoy ion counts are used to estimate the number of ions which are randomly falling into DICE windows (background noise).
7. Accumulations of ions (peaks) in RT- and *m/z*-space around the expected DICE-window per IceR feature and sample are detected by normal KDE (function kde2d in R-package MASS, version 7.3–51.5) and subsequent 2D-peak detection (local maxima). By default, KDE is performed with a resolution (grid points per dimension) of 50 and locations of up to 5 peaks (sorted by distance from expected peak location) with at least *n* detected ions (*n* = median decoy ion count) or localized within expected DICE-window are stored. Increasing KDE resolution improves the resolving power to detect peak locations, however, comes at the cost of longer processing times (doubling the resolution in theory results in quadruplicated processing times).
8. For every IceR feature, a peak per sample is selected. For samples with detected MaxQ feature, the peak closest to the expected peak location is selected as long as at least n ions form the peak (*n* = median decoy ion count) and the peak is located within the expected DICE window. These peaks are called known. In all other cases, the peak closest to known peaks (in other samples), which is not overlapping with any other peak observed in samples with known peaks, is selected as long as its *m/z* is not deviating more than 3 times the *m/z* alignment window and its RT is not deviating more than the RT alignment window. If no peak is fulfilling these criteria for a sample, the expected DICE window for this IceR feature is selected. Finally, all ions within the selected DICE windows are counted and their intensities are summed. This total intensity is further distinguished into signal and background ion intensities by defining ions with an intensity higher than the background noise at respective RT (background noise GAM)+2x standard deviation of background noise (decoy feature quantifications) as signal ions.
9. The significance of ion accumulation per quantification is determined by comparing a number of observed ions in DICE windows against expected background noise ion count distributions (observed decoy feature ions). All IceR features per sample with, e.g. a quantification *p* value < 0.05 show a significantly higher accumulation of ions than expected by chance and are thus regarded as truly present. The quality of each quantification is further evaluated based on signal-to-noise ratios.
10. Peak selections are controlled and outliers removed. Two filters are applied: (1) Features showing significantly increased interquartile ranges for peak RT or peak *m/z* between samples are completely removed. (2) Features showing significantly deviating peak RT or *m/z* in individual samples are excluded. An additional filter is applied for +1-isotope IceR features by detecting outliers that show a significant deviation of peak RT or *m/z* between the monoisotopic and +1-isotope IceR features.
11. For every IceR analysis, its peak selection accuracy is estimated by performing a FDR analysis. For that purpose, 500 sequenced IceR features are randomly selected per sample, their known true peak locations are masked (treated as if no MaxQuant feature was detected) and it is then

evaluated, how often the algorithm ends up selecting a wrong peak with deviating intensity.

12. Optional: IceR feature quantifications without any ion falling into the DICE window result in missing values e.g. in case of true absence of a peptide. In this case, the decoy feature-based sample-specific background noise models can be used to impute missing values with feature-specific background noise intensities. By default, data is imputed, however, none-imputed data is processed in parallel.

13. Peptide quantifications are aggregated to protein quantifications. The user can decide between the Top3, total sum intensity and MaxLFQ approach.

IceR results are stored as tab-delimited text files.

**Data of additional quantitative workflows**. To enable comparison of IceR against DeMix-Q and IonStar at their optimal parameter settings for the respective tool data sets, published quantitative workflow results were used:

- DeMix-Q—results for the iPRG2015 dataset—available in the supplementary materials of the original publication[8]
- IonStar—results for the Shen et al. dataset—available as Dataset S2 in the original publication[12]

Similarly, for comparison of IceR against HRM-DIA published by Bruderer et al.[31], quantitative workflow results were downloaded from the supplementary materials of the original publication[31].

**Data filtering and normalization**. The following filtering criteria were applied to respective quantitative workflow outputs for data sets 1, 2, 3, 4, 6, and 7:

- Remove contaminants, reverse hits, proteins identified only by PTM peptides and proteins identified with <2 peptides.
- Keep protein quantifications per sample based on at least 2 peptides/features.

In the case of data set 5 the respective quantitative workflow outputs the following filtering criteria were applied:

- Remove contaminants, reverse hits, proteins identified only by PTM peptide and proteins identified with <2 (1 in case of DE and CellTrail analyses on MaxQuant data) peptides.
- Keep protein quantifications per sample based on at least 2 (1 in case of DE and CellTrail analyses on MaxQuant data) peptides/features.

Subsequent to data filtering, protein and peptide quantities of samples of respective quantitative workflow outputs were median-normalized. In the case of spike-in data sets, normalization factors were calculated based on the constant background proteins.

**General data analysis**. Data analyses were performed and results were visualized using R (Version 4.0.2). Missing value imputation was performed using the methods min, lls, bpca, svd, and knn (R-packages imputeLCMD V2.0 and pca-Methods V1.80.0). For differential expression analyses on protein level, a modified $t$-test (R-package limma, version 3.44.3) was applied. Differential expression analyses on peptide level were performed using peptide-level expression-change averaging (R-package PECA[46], version 1.22.0, ordinary $t$-test, ratio based on Top5 abundant features. In the case of limma and peca, missing values were allowed as long as the respective protein was at least quantified in 2 samples of each group per pairwise comparison. Significance thresholds were defined depending on the minimal spike-in ratio per data set and Benjamini–Hochberg corrected $p$ values: (1) iPRG 2015 dataset[28], adjusted pValue < 0.05, (2) Ramus et al. dataset[29], fold-change > 2-fold, adjusted $p$ value < 0.01, (3) Shen et al. dataset[12], fold-change > 1.15-fold, adjusted $p$ value < 0.05, (4) Bruderer et al. dataset[31], fold-change > 1.15-fold, adjusted $p$ value < 0.01, (5) Zhu et al. dataset[36], fold-change > 1.5-fold, adjusted $p$ value < 0.05, (6) in-house data set, fold-change > 1.15, adjusted $p$ value < 0.01. Thresholds were applied consistently for all evaluated quantification workflows. ROCs and areas under the ROC (AUROC) were utilized to compare performances of quantitative workflows for detecting differentially abundant proteins (R-package pROC, version 1.16.2).

**Single-cell analysis**. Missing values of protein quantifications in MaxQuant data were imputed by random draws from a Gaussian distribution centred to the 1%-quantile of observed values per sample (R-package imputeLCMD version 2.0). Dimensional reduction for visualization was performed using tSNE (R-package tsne, version 0.1.3). Clustering of single cells was evaluated using the Silhouette score (R-package cluster, version 2.1.0). Unsupervised de novo chronological ordering of cells was performed using the R-package CellTrails[39] (version 1.4.0).

**Reporting summary**. Further information on research design is available in the Nature Research Reporting Summary linked to this article.

## Data availability

The source data of all analyses are supplied as Supplementary Data 1–7. The mass spectrometry proteomics data generated for this study have been deposited to the ProteomeXchange Consortium via the PRIDE[45] partner repository with the dataset identifier PXD019777 and PXD021425. Protein sequences were taken from the UniProt database (https://www.uniprot.org/). Raw LC–MS/MS data of the individual publicly available data sets were downloaded from respective sources: iPRG 2015 dataset:[28] MSV000079843 [https://massive.ucsd.edu/ProteoSAFe/dataset.jsp?task=eccf4bd3e86a4f79af468b0010eb80b0] or ftp://iprg_study:ABRF329@ftp.peptideatlas.org/ Ramus et al.[29] dataset: PXD001819. Shen et al. dataset:[12] PXD003881. Bruderer et al.[31] dataset: PASS00589. Zhu et al.[36] dataset: PXD014256. Geyer et al.[33] dataset: PXD002854. Published quantitative workflow results were downloaded from respective sources: DeMix-Q—results for the iPRG2015 dataset—available in the supplementary materials of the original publication[8]. IonStar – results for the Shen et al. dataset—available as Dataset S2 in the original publication[12]. HRM-DIA—results for the Bruderer et al. dataset—available in the supplementary materials of the original publication[31].

## Code availability

An implementation of the above described IceR procedure is available as an R-package at https://github.com/mathiaskalxdorf/IceR[27].

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

## Acknowledgements

This work was supported by the German Ministry of Education and Research (BMBF), as part of the National Research Node "Mass spectrometry in Systems Medicine" (MSCoreSys), under grant agreement 031L0212A.

## Author contributions

M.K. developed IceR; M.K. and T.M. designed experiments; T.M. performed experiments; M.K. analysed data; M.K. and J.K. wrote the manuscript; O.S. and J.K. supervised the work.

## Funding

## Competing interests

The authors declare no competing interests.
