## [Peer Review File · Nature Communications]

REVIEWER COMMENTS

Reviewer #1 (Remarks to the Author):

The manuscript by Kalxdorf et al. describes an improved computational tool to extract and analyze label free proteomics data acquired by data-dependent acquisition (DDA). Main novelties are the improved fitting of a hybrid PIP approach (combining PIP feature detection and ion-based PIP) with kernel density maps. The authors benchmark their workflow with other publicly available software such as MaxQuant, DeMix-Q and IonStar. The comparison was made using published data sets of different biological complexities: Spike-in proteins, E.Coli, Blood samples and single-cells. IceR performed better compared to the other data processing workflows included in the study not only by reducing missing values and increasing quantified proteins across the sample set, but also while keeping CV at comparable if not better levels - impressive! Especially applied to the single cell data, the improvements are significant and demonstrate clear benefits of applying computational proteomics methods such as IceR.

The authors even went as far as to test their workflow on timsTOF type of acquisition and compare to DIA label free quantification. In the comparison with DIA, IceR identifies approximately the same amount of protein with a relatively similar quantification accuracy. After subjecting the workflow to timsTOF type of data the workflow increased the identified proteins across samples. It is the first open-source "match between runs" proteomics workflow available for an ion mobility instrument.

It is a well-written article and it is very exciting to see how computational analysis is able to lift the field of proteomics, especially in order to alleviate the common "missing values" problem. However, a few criteria are not yet met to be able to deem the work ready for publication, which we will elaborate on below. The main obstacle is the fact that we haven't been able to make the R package run on our platforms, either Mac or Windows, hence leaving it questionable how well the software truly works as a universal package.

Major comments

1. The extent of novelty of the method is somewhat unclear as it seems to be a combination of two existing previously existing methods. It would be beneficial if the authors would provide a comparison to similar methods such as the recently developed Proteome Discoverer node "apQuant", DART-ID or other well-established methods in the field such as MSFragger. The fact that they outperform MaxQuant is promising, but not exactly a true test of the power of IceR compared to similar computational workflows. In terms of isobaric labeling, it would be of great interest to compare their results to similar efforts published by Yu et al. J. Proteome Res. 2020 for labeled samples.

2. The execution of the IceR package has been troublesome in our trials, resulting in us not having been successful in deploying the method on some actual data. We tested both on Mac and Windows 10, and while it installs and executes on Windows, it does not manage to complete any analysis (Error 1: Warning: Error in if: argument is of length zero. Error 2: Warning: Error in setwd: cannot change working directory)

On Mac, the package does not install correctly despite all dependencies being met (Error: objects 'choose.dir', 'setWinProgressBar', 'winProgressBar' are not exported by 'namespace:utils'
Execution halted

ERROR: lazy loading failed for package 'IceR'

* removing '/Library/Frameworks/R.framework/Versions/3.6/Resources/library/IceR'

Error: Failed to install 'IceR' from GitHub:

(converted from warning) installation of package

'/tmp/RtmploQH11/filed10864a1d82c/IceR_1.0.0.tar.gz' had non-zero exit status). It might be a consideration to generate a Docker image or similar to overcome potential compatibility issues.

3. Related to point 2, given that the authors are focusing on low sample input proteomics, I strongly believe they cannot overlook FAIMS-based data and need to support such data/raw files easily.

Therefore, they should consider adding Proteome Discoverer support as having to extract multi-CV raw files into separate sub-files, converting to mzXML and processing them through MaxQuant separately is rather tedious. It would really lift the universal implementation in labs across the world if this could be implemented, and with PD, it would even provide scope for providing an integrated analysis node, yet again expanding the audience of their efforts.

Code specific comments:

1. Overfitting and co-isolating ions may cause a significant increase in the single cell protein coverage.

In the code (lines 121-123), does the the hybrid feature work in an “IF” functional parameter or an “OR” functional parameter? In the file: “Feature_alignment_requantification.R” it has the following lines:

```
2907 ####now search for potential fitting MaxQ features, if MaxQ feature mass fits to potential peptide
and more than one species (e.g. two and three charges)
2908 ####were detected, take this charge for which more MaxQ features were found
2909 load("Temporary_files\\Feature_alignment_QC_data.RData")
2910 borders_RT <- QC_data$Feature_alignment_windows$RT_window
2911 borders_m.z <- QC_data$Feature_alignment_windows$mz_window
```

Is it an indication that the hybrid alignment will always take a feature-that is better aligned (feature-based PIP “OR” ion-based PIP).

2. Is the following FDR cleanup excluding the possible false positive hits or just the noise?

```
5061 ##Now predict background intensities per sample and feature by using individual-ly fitted GAMs,
multiply with median decoy ion count and add some noise using observed decoy intensity sds per
sample
```

```
5062 background_intensity_GAM_table_per_feature <- NULL ####used for determining which ions are
background ions and which are signal ions
```

```
5063 background_intensity_GAM_table_per_feature_sum <- NULL ####used to later impute missing
values
```

3. The indicated kernel density estimation maps are applied after the PIP fitting and the FDR estimations are applied after such a correction. Wouldn't that hide more False Positives?

4. Random Forest models are trained using 80% of the data, and validated using the remaining 20%; are there any round-robin iterations being done to avoid over-training and -fitting of the data?

Minor comments

1. Correct, but rather bold claims in the introduction regarding the lack of utility for DICE and IonStar. These statements should be rephrased or excluded.

2. LINE121- Do you also quantify using feature based PIP or only identify? It is not clear in the text before we reach the quantification using Spike-In proteins.

3. LINE 123- Why do you first explain the steps 6-8 and then step 5?

4. LINE 314-316 – Would be preferable to show peptide quantity instead of feature quantity. You can indicate in the brackets (on average) features per peptide.

5. Figure 4C, what is meant by the 0% in the lower half of the bar charts?

Reviewer #2 (Remarks to the Author):

Kalxdorf et al. present a new R package IceR for label-free DDA-based quantitative mass spectrometry proteomics. The tool implements a combination of feature-based and ion-based peptide identity propagation to reduce the number of missing values in label-free DDA proteomics data.

Overall, the package provides a collection of previously introduced techniques to process DDA proteomics data. While it may be a useful resource for people working with open source tools for proteomics data analysis, the approach itself is not novel. Similar approaches have been implemented in both open-source (e.g. DeMix-Q, IonStar) and commercial (e.g. Progenesis) software.

When introducing a new tool, it is important to systematically compare its performance to the relevant available tools. Now, most of the comparisons are made only against MaxQuant, while no systematic comparisons have been made against the previously introduced DeMix-Q and IonStar methods. Such comparisons should be included. Additionally, it would be nice to see, how the commercial software Progenesis would work in the datasets. At least, it would perhaps be worth discussing in the manuscript.

Overall, acquiring intensities for non-identified features relies heavily on the algorithms that perform LC-MS feature alignment, for which several groups have developed algorithms throughout the years (e.g. DOI: 10.1093/bib/bbt080 for a comprehensive list). The proposed IceR alignment and peptide identification propagation should therefore be presented in the larger context, instead of simply extending the complete MaxQuant workflow.

As a reference MaxQuant method, the authors mention that the MaxLFQ algorithm was used for quantification. Sometimes, however, it seems that the number of missing values is greatly increased when using the MaxLFQ values instead of using the iBAQ values from MaxQuant and preprocessing them separately (e.g. DOI: 10.1016/j.jprot.2020.103669). While the effect seems to be dataset specific, it would be important to understand the impact of this choice on the performance of MaxQuant. Also, would this choice affect the performance of IceR itself?

As mentioned by the authors, imputation has been suggested as an alternative approach to tackle missing values in DDA proteomics data. However, it has been shown that the choice of the imputation strategy may have a major impact on the results and a properly chosen imputation strategy can greatly improve the performance of MaxQuant (e.g. DOI: 10.1093/bib/bbx054). In general, the best strategies for imputation are not zero or background imputation (similar to the random draws from the lowest 1% of values used by the authors here). Therefore, it would be interesting to see, how imputing MaxQuant with a different imputation strategy (e.g. LLS) would compare to IceR.

The IceR approach itself includes a total of 13 sequential steps. It is not clear, what is the importance of each step and what are the most critical steps in terms of the performance? Are all the steps really needed? Moreover, there seem to be a lot of parameters involved in the different steps. What is their role in the performance? How much parameter tuning is required for IceR and what parameters were used in the present analyses? How about MaxQuant?

I see the IceR software package implementation as the main contribution of this work. Currently, however, the package falls short of its promise (IceR 1.0.0, GitHub). Despite having a Shiny interface, the package appears to rely on Windows specific features (e.g. progress bar and folder selection) thus greatly limiting its usage, for example, in HPC environments, where Linux is often favoured. This means that the statement "tested on Windows 10 but should work on other OS as well" falls short. The authors should thoroughly test their package on all operating systems. The IceR package could also benefit from being included in a public and commonly used repository, such as CRAN or Bioconductor. Even if not, the software package should be polished and pass the appropriate checks (R CMD check, or equivalent) without errors or warnings. For instance, currently, the software relies on dozens of other R packages and there are numerous warnings related to IceR replacing existing functions, which leads to potential issues when the software is used.

Minor comments

The authors could have provided more information for the reader to understand the rationale of their choices.

The ROC curves in each dataset always combine the results from all the pairwise comparisons. It would be interesting to see the results separately for each comparison. More details would also be needed regarding the treatment of missing values in the comparisons. For statistical analysis, did you allow any missing values?

The use of the term "sequencing" is a bit confusing.

On page 2, lines 95-96, the authors say that "It [IceR] can be seamlessly integrated with the MaxQuant suite, or with any other pre-processed label-free proteomics data set for which detected features are reported". However, the IceR software expects MaxQuant output and all benchmarks in the manuscript are carried out only for MaxQuant output. While MaxQuant is a popular non-commercial software, it would be valuable to combine IceR with other software, such as OpenMS for the construction of more automated workflows, which would increase its value and usability.

Reviewer #1 (Remarks to the Author):

The manuscript by Kalxdorf et al. describes an improved computational tool to extract and analyze label free proteomics data acquired by data-dependent acquisition (DDA). Main novelties are the improved fitting of a hybrid PIP approach (combining PIP feature detection and ion-based PIP) with kernel density maps. The authors benchmark their workflow with other publicly available software such as MaxQuant, DeMix-Q and IonStar. The comparison was made using published data sets of different biological complexities: Spike-in proteins, E.Coli, Blood samples and single-cells. IceR performed better compared to the other data processing workflows included in the study not only by reducing missing values and increasing quantified proteins across the sample set, but also while keeping CV at comparable if not better levels - impressive! Especially applied to the single cell data, the improvements are significant and demonstrate clear benefits of applying computational proteomics methods such as IceR.

The authors even went as far as to test their workflow on timsTOF type of acquisition and compare to DIA label free quantification. In the comparison with DIA, IceR identifies approximately the same amount of protein with a relatively similar quantification accuracy. After subjecting the workflow to timsTOF type of data the workflow increased the identified proteins across samples. It is the first open-source "match between runs" proteomics workflow available for an ion mobility instrument.

It is a well-written article and it is very exciting to see how computational analysis is able to lift the field of proteomics, especially in order to alleviate the common "missing values" problem.

We thank the reviewer for his/her enthusiastic comments to our manuscript.

However, a few criteria are not yet met to be able to deem the work ready for publication, which we will elaborate on below. The main obstacle is the fact that we haven't been able to make the R package run on our platforms, either Mac or Windows, hence leaving it questionable how well the software truly works as a universal package.

Major comments

1. The extent of novelty of the method is somewhat unclear as it seems to be a combination of two existing previously existing methods. It would be beneficial if the authors would provide a comparison to similar methods such as the recently developed Proteome Discoverer node "apQuant", DART-ID or other well-established methods in the field such as MSFragger. The fact that they outperform MaxQuant is promising, but not exactly a true test of the power of IceR compared to similar computational workflows. In terms of isobaric labeling, it would be of great interest to compare their results to similar efforts published by Yu et al. J. Proteome Res. 2020 for labeled samples.

Thanks for pointing this out. We could have done a better job to clarify the novelty of our approach.

Proteomics quantification workflows typically perform feature-based peptide identity propagation (PIP) enabling the assignment of a peptide identifier to a feature even if it had not been selected for fragmentation. Example workflows are MaxQuant, Proteome Discoverer with apQuant, and MSFragger with IonQuant. However, feature-based PIP requires isotope peak patterns to be recognized as features, which may fail in case of low-abundance features, thereby prohibiting PIP, thus limiting the sensitivity of this approach. In contrast, ion-based PIP solely relies on matching of m/z and RT (and IM) windows using Direct Ion Current Extraction (DICE) independent of feature-detection

algorithms, allowing more sensitive identity propagation compared to feature-based PIP. Despite these key advantages of ion-based PIP and DICE, they have been used only in very few approaches (e.g. DeMix-Q or IonStar), which have remained niche-applications in proteomics mainly because of difficulties in running these tools, especially for larger data sets. Therefore we developed IceR as a completely new tool, implementing DICE with ion-based and feature-based PIP to combine the best of these approaches. More specifically, IceR has 4 unique key features:

1. On top of a global chromatographic retention time alignment, which minimizes chromatographic variability between samples by correcting deviations using fitted models, IceR additionally performs for each individual feature a statistically-controlled, local, and sample-specific peak alignment using kernel density estimations. This second step is important to correct for additional non-systematic and sample-specific deviations in m/z, RT, and IM-direction. This additional step allows much narrower and feature-specific DICE-windows in m/z-, RT- and IM-space compared to the fixed and large DICE-windows (e.g. $m/z \pm 5$ ppm, $RT \pm 1$ min) applied by DeMix-Q and IonStar.
2. Already the occurrence of a single ion within the DICE-window of a feature can be used to estimate the abundance of the respective peptide, by applying stringent statistical thresholding. The latter is implemented to account for potential noise signals in the quantification windows that can be falsely interpreted as the presence of a peptide. IceR statistically scores every feature quantification based on the observed accumulation of ions within DICE windows, thereby allowing the distinction between true signal and background noise. This metric is not available in DeMix-Q or IonStar.
3. IceR implements a hybrid PIP approach, combining feature-based and ion-based PIP. MaxQuant is a commonly applied quantification workflow in the proteomics community due to its simple and free usage. However, it only enables feature-based PIP with the above-described drawbacks. On the other hand, tools that allow usage of DICE and ion-based PIP (like DeMix-Q and IonStar) demonstrate highly reduced missing value rates and improved quantification accuracies. IceR combines the best of both strategies by taking a hybrid PIP approach, supplementing feature-based PIP with the sensitivity of ion-based PIP to rescue low abundant features: IceR will first perform a feature-based PIP (match between runs, like implemented in MaxQuant), and only when a suitable matching feature is absent, the ion-based PIP will be applied. In this case, IceR looks for accumulations of ions (peaks) within the expected m/z and RT (+ IM) space, and finally picks the best matching peak if several quality criteria are met (no significant deviation in RT from all other samples, no significant deviation in m/z from all other samples, significant accumulation of ions within DICE window). This is all implemented in a unified R tool for easy adoption in the field.
4. Unlike DeMix-Q and IonStar, IceR allows inference of protein abundances from feature quantifications using the MaxLFQ algorithm which is known to perform better than summing or TopN approaches (e.g. DOI: 10.1016/j.jprot.2020.103669).

In addition, IceR comes free of charge for academic and industrial environments unlike Proteome Discoverer, MSFragger, and IonStar.

We revised the respective paragraph in the manuscript to better highlight the novelty of IceR in comparison to established workflows.

To allow a more comprehensive evaluation of the outstanding performance of IceR, we have followed the reviewer's suggestion and have compared IceR to apQuant and MSFragger for all relevant spike-in datasets (Fig. 1, Fig.2 and Fig. 5). After re-analyzing these data, this is now indicated in **updated Fig 1C-D-E-F, Fig 2B-C-D, Fig 5A-B-C-D-E, Suppl Fig 4A-N, Suppl Fig 5, Suppl Fig 6, Suppl Fig 9A-N, Suppl Fig 10**). In brief, for all datasets, IceR resulted in best sensitivity at lowest false discovery rates especially

when performing differential abundance testing on peptide-level using peca. Missing value imputation could improve sensitivity of the other tools, however, at the cost of generally highly increased false discovery rates.

2. The execution of the IceR package has been troublesome in our trials, resulting in us not having been successful in deploying the method on some actual data. We tested both on Mac and Windows 10, and while it installs and executes on Windows, it does not manage to complete any analysis (Error 1: Warning: Error in if: argument is of length zero. Error 2: Warning: Error in setwd: cannot change working directory)

On Mac, the package does not install correctly despite all dependencies being met (Error: objects 'choose.dir', 'setWinProgressBar', 'winProgressBar' are not exported by 'namespace:utils')

Execution halted

ERROR: lazy loading failed for package 'IceR'

* removing '/Library/Frameworks/R.framework/Versions/3.6/Resources/library/IceR'

Error: Failed to install 'IceR' from GitHub:

(converted from warning) installation of package

'/tmp/RtmploQHu1/filed10864a1d82c/IceR_1.0.0.tar.gz' had non-zero exit status). It might be a consideration to generate a Docker image or similar to overcome potential compatibility issues.

Thanks for pointing this out, obviously this was a clear omission. IceR was originally only tested for Windows 10 and it also used several Windows-specific R functions. We have now corrected this, optimized code to pass all quality criteria that are required by the R cmd check, and we now successfully tested IceR on multiple devices, on multiple operating systems (Win10, Ubuntu, macOS) and by several people. Based on the reported error log by the reviewer we have the impression that the GUI of IceR was not used (consider running R as administrator/superuser) as e.g. described for the example data set on GitHub. Furthermore, it is important to keep in mind that automated conversion of Thermo Raw files into mzXML by the ProteoWizard tool msconvert is currently only available for Windows. Hence in Linux and macOS this conversion has to be triggered manually.

3. Related to point 2, given that the authors are focusing on low sample input proteomics, I strongly believe they cannot overlook FAIMS-based data and need to support such data/raw files easily. Therefore, they should consider adding Proteome Discoverer support as having to extract multi-CV raw files into separate sub-files, converting to mzXML and processing them through MaxQuant separately is rather tedious. It would really lift the universal implementation in labs across the world if this could be implemented, and with PD, it would even provide scope for providing an integrated analysis node, yet again expanding the audience of their efforts.

Thanks for this suggestion. While it is true that the support for FAIMS-based data could be highly beneficial, many tools still struggle to support multi-CV FAIMS data, including MaxQuant. Instead of developing a work-around requiring the user to go through multiple analysis pipelines, we decided to wait until MaxQuant natively supports FAIMS data. Furthermore, IceR is currently designed with seamless integration into the MaxQuant analysis workflow. Proteome Discoverer is a commercial tool and hence is currently out of focus. Furthermore, enabling use of IceR for additional other preprocessing pipelines would require a lot of additional effort which is currently beyond the scope of this work. However, future versions of IceR will include native support for other preprocessing pipelines.

Code specific comments:

1. Overfitting and co-isolating ions may cause a significant increase in the single cell protein coverage.

In the code (lines 121-123), does the the hybrid feature work in an “IF” functional parameter or an “OR” functional parameter? In the file: “Feature_alignment_requantification.R” it has the following lines:

```
2907 ####now search for potential fitting MaxQ features, if MaxQ feature mass fits to potential
peptide and more than one species (e.g. two and three charges)
2908 ####were detected, take this charge for which more MaxQ features were found
2909 load("Temporary_files\\Feature_alignment_QC_data.RData")
2910 borders_RT <- QC_data$Feature_alignment_windows$RT_window
2911 borders_m.z <- QC_data$Feature_alignment_windows$mz_window
```

Is it an indication that the hybrid alignment will always take a feature-that is better aligned (feature-based PIP “OR” ion-based PIP).

This section of code is originating from a currently experimental but unused function in IceR. It was intended to allow for searching MS1 features that could fit to completely missing peptides of identified proteins. However, as stated above, this is an experimental code section and to avoid confusion, we now removed this function from the IceR package.

Still, to answer the question: The hybrid PIP works in an “or” approach, first performing a feature-based PIP (match between runs like implemented in MaxQ) and only invoking ion-based PIP when a suitable matching feature is absent (see our response to point 1 above for a more detailed explanation).

2. Is the following FDR cleanup excluding the possible false positive hits or just the noise?

```
5061 ####Now predict background intensities per sample and feature by using individual-ly fitted
GAMs, multiply with median decoy ion count and add some noise using observed decoy intensity sds
per sample
5062 background_intensity_GAM_table_per_feature <- NULL ####used for determining which ions
are background ions and which are signal ions
5063 background_intensity_GAM_table_per_feature_sum <- NULL ####used to later impute missing
values
```

Based on the introduced decoy features, IceR estimates the number and intensity of ions which fall into the used DICE-windows by chance. Generalized additive models (GAMs) are fitted to the data to predict the intensity of noise as a function of the chromatographic retention time per sample. Using these models, every observed ion within the DICE-window is categorized to be either a signal ion ($\text{intensity} > \text{GAM}(\text{rt}) + 2x \text{sd}$) or background ion ($\text{intensity} < \text{GAM}(\text{rt}) + 2x \text{sd}$). This allows estimating the signal to noise ratio per feature and sample. Furthermore, based on the observed number of ions in each DICE window, a p-value is calculated for each feature quantification to indicate whether more ions were observed in the DICE window than was expected by chance. Both quality criteria are reported for every feature quantification and can be used for evaluation of quantification quality. However, to allow highest possible sensitivity, by default quantifications are currently not filtered based on these metrics. Hence, as a shorter answer to the reviewer’s question, the noise FDR is calculated and reported, however no data are excluded.

3. The indicated kernel density estimation maps are applied after the PIP fitting and the FDR estimations are applied after such a correction. Wouldn't that hide more False Positives?

After a first global modelling-based alignment of RT and m/z (and IM) between samples, PIP is performed in two steps:

- 1) Identity of features is shared across samples by feature-based PIP only for features with high matching confidence (narrow RT and m/z windows).
- 2) To allow PIP for remaining samples for which no feature could be matched in 1), kernel density maps are used for further reduction of m/z and RT (and IM) deviations between samples, and finally ion-based PIP is used for identity propagation.

Subsequently, the quality of every peak selection is estimated by several quality measures:

- 3) Significance of quantification per feature by comparing the number of observed ion counts to modelled decoy-feature ion counts
- 4) Quality of quantification per feature by comparing ion intensities to modelled decoy-feature ion intensities
- 5) Confidence of selected peaks per feature by comparing observed m/z and RT (and IM) per sample against all other samples

Measures 3 and 4 are solely used for estimating quantification quality but not for filtering PIP. However, measure 5 is used to filter PIPs that resulted in the selection of peaks that highly deviated from observed m/z and/or RT (and IM) in all other samples in which this peptide was identified (by MS/MS).

Finally, with all this information from the above-described steps, IceR estimates per sample how often it would pick a wrong peak if it would have to use ion-based PIP to find the respective feature. This is done by random selection of 500 truly identified features (by MS/MS) per sample, mask their identity, and re-identify them by ion-based PIP. Typically, this results in a peak-selection FDR < 1 %.

As peak selection FDR is solely estimated based on truly identified features, false positives are not hidden by the above-described approach, and all applied steps only help to reduce false positives.

4. Random Forest models are trained using 80% of the data, and validated using the remaining 20%; are there any round-robin iterations being done to avoid over-training and -fitting of the data?

Thank you for this comment. It is correct that a cross-validation strategy using round-robin iterations and multiple folds could be employed to evaluate the model performance. We did not do this, as the main value is diagnostics and hence a single split (fold) is sufficient. We would like to stress that no model parameters are optimized on the test set and hence the assessment of the model, which is trained individually for each sample, is not affected by overfitting. For each sample, the data (features) are split into an 80 % training fraction and 20 % fraction for validation. With this approach overfitting could be still detected as it would result in poor correlation coefficients in case of the 20 % validation data that deviates from the performance on the training fraction. However, this was so far not observed in any of the data sets.

We agree that in order to assess the generalization performance across samples, a more sophisticated cross validation scheme could be employed. We did not do this in our study, primarily because the goal of this analysis is diagnostic to estimate the expected m/z correction factors for features which were missed within the respective sample. Consequently, a generalization across samples is not intended

for these models. Furthermore, these models are only used for a first rough estimation of required m/z correction factors as the final corrections are performed by IceR using kernel density estimated ion accumulation maps (steps 6-8 of the workflow, visualized in Supplementary Fig. 1). Hence, again stressing the diagnostic utility and use of these models.

Minor comments

1. Correct, but rather bold claims in the introduction regarding the lack of utility for DICE and IonStar. These statements should be rephrased or excluded.

We assume that the reviewer refers to the following: 'In addition, running of DeMix-Q and IonStar is highly cumbersome, requiring installation of several tools including discontinued commercial applications. This may explain why DICE-based approaches, despite their advantages, have never permeated into mainstream proteomics applications.'

We have now changed this into: 'In addition, running of DeMix-Q and IonStar is not straightforward, requiring installation of several tools including discontinued commercial applications. This may explain why DICE-based approaches, despite their clear advantages, have not permeated into mainstream proteomics applications.'

2. LINE121- Do you also quantify using feature based PIP or only identify? It is not clear in the text before we reach the quantification using Spike-In proteins.

Thanks for pointing us to this unclear phrasing. The PIP approaches are solely used for identity propagation. Quantification is performed in both cases by DICE. We rephrased the sentence to:

"Next, IceR incorporates a hybrid PIP approach: peptide information is propagated preferably by feature-based PIP, and identities are recovered by ion-based PIP only in cases of missing feature detection."

3. LINE 123- Why do you first explain the steps 6-8 and then step 5?

Thanks for pointing us to this issue. We rephrased the paragraph to:

"These features are aggregated and aligned over all samples (steps 1-3 in Supplementary Fig. 1), and finally the respective quantities are extracted by DICE from MS raw files. To enable reliable PIP, IceR performs modelling-based RT- and m/z-corrections (step 4), similar to MaxQuant, DeMix-Q and IonStar. Next, IceR incorporates a hybrid PIP approach: peptide information is propagated preferably by feature-based PIP, and identities are recovered by ion-based PIP only in cases of missing feature detection (step 5). To further improve reliability of ion-based PIP, IceR performs an additional alignment step. While step 4 can greatly decrease global variability between samples, local sample- and feature-specific heterogeneities might still be missed, which could result in false matches. To account for this, IceR uniquely performs a second sample- and feature-specific alignment step based on kernel density estimated ion accumulation maps (steps 6-8)."

4. LINE 314-316 – Would be preferable to show peptide quantity instead of feature quantity. You can indicate in the brackets (on average) features per peptide.

Thanks for this comment. We now state the number of quantified peptides instead of features per single cell sample and adjusted Fig. 7b accordingly.

5. Figure 4C, what is meant by the 0% in the lower half of the bar charts?

Fig. 4c (similar to Fig. 4f) illustrates the absolute number of observed true positives (upper bars) and number of observed false positives (lower bars) across all (28) pairwise differential expression analyses (cumulative). The corresponding true positive rates (observed true positives vs expected true positives) and false positive rates (observed false positives vs expected true negatives) are indicated as relative fractions within the bars. All 3 methods resulted in false positive rates < 0.1 % (hence 0.0%). To clarify this, we rephrased the corresponding figure legend for Fig. 4c (and Fig. 4f) to:

“Cumulative true and false positive counts over all (28) pairwise DE analyses for MaxQuant (grey), DIA (dark grey) and IceR (orange). Percentages within bars indicate corresponding true positive and false positive rates.”

Reviewer #2 (Remarks to the Author):

Kalxdorf et al. present a new R package IceR for label-free DDA-based quantitative mass spectrometry proteomics. The tool implements a combination of feature-based and ion-based peptide identity propagation to reduce the number of missing values in label-free DDA proteomics data. Overall, the package provides a collection of previously introduced techniques to process DDA proteomics data. While it may be a useful resource for people working with open source tools for proteomics data analysis, the approach itself is not novel. Similar approaches have been implemented in both open-source (e.g. DeMix-Q, IonStar) and commercial (e.g. Progenesis) software.

Thanks for pointing this out. We could have done a better job to clarify the novelty of our approach.

However, first it is important to keep in mind that only DeMix-Q comes free of charge as IonStar requires SIEVE, a discontinued commercial Software from Thermo Scientific. Furthermore, due to the discontinued support of SIEVE, we have not been able to get IonStar running. Similarly, even with the support from the authors of DeMix-Q, we could not get this tool running in our hands due to installation issues. These issues are most likely due to incompatibilities with updated versions of associated/required tools by DeMix-Q, as authors are no longer supporting/updating DeMix-Q (published 5 years ago). On the other hand, Progenesis QI is a commercial tool and hence might have a smaller user base. Unfortunately, we have no access to this tool, however a recent comprehensive comparison of popular proteomics software workflows showed that Progenesis QI identified a considerably lower numbers of proteins compared to e.g. MaxQuant, despite good performance in detecting true positives in spike-in data sets on relative scale (ROC curves, pAUC), (<https://doi.org/10.1093/bib/bbx054>). Furthermore, this paper showed that applying missing value imputation methods had beneficial effects for all tools except Progenesis QI letting it even fall behind MaxQuant when looking on relative performance metrics (pAUC).

Apart from this, we like to emphasize that IceR is distinct from any of the mentioned tools, going far beyond the capabilities of DeMix-Q and IonStar (and Progenesis QI). Despite the fact that all 3 tools apply ion-based PIP for improved data completeness and quantification accuracy, there are 4 key differences that distinguish IceR from these tools:

1. On top of a global chromatographic retention time alignment, which minimizes chromatographic variability between samples by correcting deviations using fitted models, IceR additionally performs for each individual feature a statistically-controlled, local, and sample-specific peak alignment using kernel density estimations. This second step is important to correct for additional non-systematic and sample-specific deviations in m/z, RT, and IM-direction. This additional step allows much narrower and feature-specific DICE-windows in m/z-, RT- and IM-space compared to the fixed and large DICE-windows (e.g. $m/z \pm 5$ ppm, $RT \pm 1$ min) applied by DeMix-Q and IonStar.
2. Already the occurrence of a single ion within the DICE-window of a feature can be used to estimate the abundance of the respective peptide, by applying stringent statistical thresholding. The latter is implemented to account for potential noise signals in the quantification windows that can be falsely interpreted as the presence of a peptide. IceR statistically scores every feature quantification based on the observed accumulation of ions within DICE windows, thereby allowing the distinction between true signal and background noise. This metric is not available in DeMix-Q or IonStar.
3. IceR implements a hybrid PIP approach, combining feature-based and ion-based PIP. MaxQuant is a commonly applied quantification workflow in the proteomics community due to its simple and free usage. However, it only enables feature-based PIP with the above-described drawbacks. On the other hand, tools that allow usage of DICE and ion-based PIP (like

DeMix-Q and IonStar) demonstrate highly reduced missing value rates and improved quantification accuracies. IceR combines the best of both strategies by taking a hybrid PIP approach, supplementing feature-based PIP with the sensitivity of ion-based PIP to rescue low abundant features: IceR will first perform a feature based PIP (match between runs, like implemented in MaxQuant), and only when a suitable matching feature is absent, the ion-based PIP will be applied. In this case, IceR looks for accumulations of ions (peaks) within the expected m/z and RT (+ IM) space, and finally picks the best matching peak if several quality criteria are met (no significant deviation in RT from all other samples, no significant deviation in m/z from all other samples, significant accumulation of ions within DICE window). This is all implemented in a unified R tool for easy adoption in the field.

4. Unlike DeMix-Q and IonStar, IceR allows inference of protein abundances from feature quantifications using the MaxLFQ algorithm which is known to perform better than summing or TopN approaches (e.g. DOI: 10.1016/j.jprot.2020.103669).

We revised the respective paragraph in the manuscript to better highlight the novelty of IceR in comparison to established workflows.

To allow a more comprehensive evaluation of the outstanding performance of IceR, we have followed the reviewer's suggestion and have compared IceR to apQuant and MSFragger for all relevant spike-in datasets (Fig. 1, Fig.2 and Fig. 5). After re-analyzing these data, this is now indicated in **updated Fig 1C-D-E-F, Fig 2B-C-D, Fig 5A-B-C-D-E, Suppl Fig 4A-N, Suppl Fig 5, Suppl Fig 6, Suppl Fig 9A-N, Suppl Fig 10**). In brief, for all datasets, IceR resulted in best sensitivity at lowest false discovery rates especially when performing differential abundance testing on peptide-level using peca. Missing value imputation could improve sensitivity of the other tools, however, at the cost of generally highly increased false discovery rates.

1. When introducing a new tool, it is important to systematically compare its performance to the relevant available tools. Now, most of the comparisons are made only against MaxQuant, while no systematic comparisons have been made against the previously introduced DeMix-Q and IonStar methods. Such comparisons should be included. Additionally, it would be nice to see, how the commercial software Progenesis would work in the datasets. At least, it would perhaps be worth discussing in the manuscript.

We thank the reviewer for this comment. Indeed, we mainly focused on the comparison against MaxQuant, but in fact we had also performed this for DeMix-Q and IonStar wherever possible. The problem we faced was that, despite intense efforts, we could not get these tools working: as described above, even with support by the authors of DeMix-Q we could not get this tool running since the required input files, which have to be generated by the preceding DeMix pipeline, could not be generated due to an error in Node 8 (DeMix wrapper, feature_ms2_clone_TOPP2.py in line "element = next(spec.xmlTree)", AttributeError: 'Spectrum' object has no attribute 'xmlTree'). IonStar requires a discontinued commercial tool from Thermo Scientific (SIEVE), hence, unfortunately, this is also not usable anymore. To still be able to compare IceR to DeMix-Q and IonStar (as already shown in our first submission), we used the originally published data for both tools, and using the associated raw MS data as an input for IceR. In fact, this allowed a direct comparative assessment of a fairly large number of performance metrics (Fig 2B-C-D-E, Suppl Fig 2G-H-I, Suppl Fig 3J-K-L-M-N, Suppl Fig 4H-I-J-L-M-N, Suppl Fig 6). In all cases, IceR outperformed these methods in terms of sensitivity and specificity.

In addition, to extend the comprehensiveness of our analyses and as suggested by Reviewer 1, we have now also analyzed all relevant spike-in datasets (Fig. 1, Fig.2 and Fig. 5) by apQuant and MSFragger (with and without missing value imputation by Bayesian principal component analysis (bpca) which

performed best in comparison to all tested imputation methods). This is now indicated in **updated Fig 1C-D-E-F, Fig 2B-C-D, Fig 5A-B-C-D-E, Suppl Fig 4A-N, Suppl Fig 5, Suppl Fig 6, Suppl Fig 9A-N, Suppl Fig 10**). In brief, for all datasets, IceR resulted in best sensitivity at lowest false discovery rates especially when performing differential abundance testing on peptide-level using peca. Missing value imputation could improve sensitivity of the other tools, however, at the cost of generally highly increased false discovery rates.

Progenesis Q1 could not be added to this comparison as it is a commercial tool that we do not have access to. Nevertheless, a recent thorough comparison of label-free quantification tools (<https://doi.org/10.1093/bib/bbx054>), indicated that Progenesis Q1 at best performs similar to MaxQuant with imputation enabled. Of note, IceR clearly outperforms MaxQuant under these conditions (e.g. Fig. 5b-e).

2. Overall, acquiring intensities for non-identified features relies heavily on the algorithms that perform LC-MS feature alignment, for which several groups have developed algorithms throughout the years (e.g. DOI: 10.1093/bib/bbt080 for a comprehensive list). The proposed IceR alignment and peptide identification propagation should therefore be presented in the larger context, instead of simply extending the complete MaxQuant workflow.

We thank the reviewer for pointing us towards this highly informative review and for the suggestion to put our new approach in a broader context. The review lists, among others, the following 8 major limitations of current LC-MS feature alignment methods:

1. Most methods simplify the alignment procedure by considering only certain dimensions of the data, e.g. total ion chromatograms (TIC) or extracted ion chromatograms (XIC). (e.g. AMSRPM, COW, DTW)
2. Most methods solely focus on correcting systematic deviations between samples by fitting warping models, thereby ignoring sample- and feature-specific deviations. (e.g. DTW, COW, PARS)
3. Random mass errors e.g. introduced by changes in electric fields, space-charge effects or temperature variability, are typically not considered during feature alignment.
4. Warping methods typically deform the original signal, which is detrimental to comparative/differential analysis.
5. Many methods incorporate intensity signals into the alignment procedure rendering them vulnerable to be biased towards high-intensity signals (e.g. Christin et al., PTW)
6. Many methods are highly complex resulting in impractical high run times (e.g. PEPPER, AMSRPM)
7. All methods except CPM and XCMS require specification of a reference sample to which all other samples are corrected thereby making results heavily dependent on yet another user-defined parameter.
8. Most methods are limited to alignment of RT and/or m/z dimension, thereby not taking full advantage of additional ion mobility data as provided by timsTOF Pro instruments.

IceR addresses all of these points by:

1+2+3 Applying a XIC-based alignment of samples using non-linear model fitting combined with spectra-based peak matching using kernel density maps. This unique 2-step approach thereby enables usage not only of XIC but also of all spectra data for the alignment process. Furthermore, it considers sample- and feature-specific random deviations not only in RT but also in m/z and IM space.

4. Re-quantifying features across all samples by DICE using identical feature-specific window sizes after alignment.
5. Incorporating ion accumulations (instead of signal intensities), estimated by kernel density maps, into the alignment process.
6. Enabling short run times on commonly equipped computers like a laptop.
7. Using median feature profiles instead of selecting profiles of a user-defined reference sample for estimation of alignment parameters.
8. Taking full advantage of m/z, RT, and IM data for feature alignment.

We added a paragraph to the introduction and discussion section to present the here described 2-step alignment approach in the larger context:

Introduction:

“Yet, ion-based PIP is even more dependent on accurate sample alignment to enable narrow DICE windows as otherwise co-eluting species can distort quantity estimations or introduce false positives.

To date, dozens of alignment algorithms for LC-MS data have been proposed of which the majority relies on fitting of warping models relative to a reference sample. While these approaches allow the correction of systematic deviations between samples, local sample- and feature-specific deviations are often overseen. Furthermore, the majority of algorithms simplify the alignment procedure by considering only certain dimensions of the data e.g. total ion chromatograms (TIC) or extracted ion chromatograms (XIC). Additionally, random mass errors occurring in the mass spectrometer e.g. caused by changes in electric fields, space-charge effects or temperature are typically ignored. To enable concise DICE windows, the choice of an alignment algorithm that enables systematic as well as local sample- and feature-specific corrections in RT but also in m/z dimension is highly important, but remains to be established.

Two recent proteomics tools that implement ion-based PIP are DeMix-Q8 and IonStar12, both achieving highly reduced missing value rates and improved sensitivity to detect differentially abundant proteins compared to MaxQuant. This shows the potential of DICE, yet a number of fundamental and practical issues remain. For instance, both DeMix-Q and IonStar require large and fixed DICE windows (e.g. defaulting to $m/z \pm 5$ ppm, $RT \pm 1$ min) resulting in data deterioration by co-eluting interferences, probably caused by sub-optimal sample alignment. Both DeMix-Q and IonStar use sophisticated alignment algorithms (based on the OpenMS proteomics pipeline and ChromAlign implemented in the commercial tool SIEVE from Thermo Scientific, respectively), however they only correct for systematic deviations, thereby ignoring potential random local feature specific deviations. Furthermore, both tools are designed to only process RT and m/z data from Thermo Fisher Scientific Orbitrap mass spectrometers, excluding their use to other vendors and scan modes (e.g. ion mobility (IM) separation). In addition, running of DeMix-Q and IonStar is not straightforward, requiring installation of several tools including discontinued commercial applications. This may explain why DICE-based approaches, despite their clear advantages, have not permeated into mainstream proteomics applications.”

Discussion:

“IceR rests on two key characteristics: 1) it combines feature-based and ion-based PIP, leveraging the strengths of both principles for robust and sensitive PIP; 2) it introduces a novel feature alignment approach that encompasses various measures. For instance, this includes correction for both systematic and local deviations such as sample- and feature-specific random variation in mass, RT and (unique for IceR) ion mobility as an additional dimension. Importantly, alignment parameters are automatically estimated from the data across all samples, instead of requiring the user to specify a single reference sample which introduces yet another dependence of results on a user-defined

parameter. Finally, consistent quantification of features across all samples is achieved by DICE using the same feature-specific window sizes instead of depending on signal intensities that are deformed by the used warping function. Many of these aspects are not taken into account in most commonly used alignment procedures, however they critically contribute to the performance of IceR. In contrast to previous DICE-based tools, IceR is offered as a comprehensive yet user-friendly R-package with an intuitive graphical user interface (Supplementary Fig. 12) that requires minimal input from the user. To illustrate this, it is worth mentioning that default settings were used for the analysis of all described data sets, showing that robust performance of IceR can be achieved across a wide range of instruments (various orbitraps and timsTOF-Pro), LC gradients (20-180 min) and sample types (plasma, bulk cell lysates, single cells).”

3. As a reference MaxQuant method, the authors mention that the MaxLFQ algorithm was used for quantification. Sometimes, however, it seems that the number of missing values is greatly increased when using the MaxLFQ values instead of using the iBAQ values from MaxQuant and preprocessing them separately (e.g. DOI: 10.1016/j.jprot.2020.103669). While the effect seems to be dataset specific, it would be important to understand the impact of this choice on the performance of MaxQuant. Also, would this choice affect the performance of IceR itself?

Many thanks for this comment. The publication of Zhao et al. describes the issue that, although MaxLFQ-based quantification results overall in most accurate and reliable quantifications, it shows higher missing values especially in case of low-abundance proteins. Due to a lack of detail in the description of the used parameters for the individual quantification methods in this publication, it is difficult to recapitulate the observed major differences in quantified proteins between e.g. iBAQ and MaxLFQ. However, assuming the authors used the default parameters in MaxQuant, the higher number of missing values in case of MaxLFQ, especially in case of low abundant proteins, could be explained by the fact that iBAQ and MaxLFQ have different requirements:

- 1) in case of iBAQ, per sample a protein only requires at least 2 quantified peptides.
- 2) In case of MaxLFQ, per sample a protein requires at least 2 quantified peptides which have to be available for pairwise-comparisons between at least two samples.

As it becomes less likely to quantify the exact same peptides in two samples with decreasing overall protein abundance, which is required by MaxLFQ but not by iBAQ, the MaxLFQ approach can suffer from higher missing value rates compared to iBAQ in case of low abundant proteins. Although this might be an issue in case of the MaxQuant quantification results, this issue does not apply to IceR, as it results in almost complete quantification of all peptides across all samples even at low intensities allowing pairwise-comparisons for almost all peptides between all samples.

4. As mentioned by the authors, imputation has been suggested as an alternative approach to tackle missing values in DDA proteomics data. However, it has been shown that the choice of the imputation strategy may have a major impact on the results and a properly chosen imputation strategy can greatly improve the performance of MaxQuant (e.g. DOI: 10.1093/bib/bbx054). In general, the best strategies for imputation are not zero or background imputation (similar to the random draws from the lowest 1% of values used by the authors here). Therefore, it would be interesting to see, how imputing MaxQuant with a different imputation strategy (e.g. LLS) would compare to IceR.

Many thanks for this great suggestion. We have now added for all relevant data sets a comparison of 5 commonly used imputation methods (random draws from the lowest 1 % (min), local least squares (lls), Bayesian principal component analysis (bPCA), singular value decomposition (svd), k-nearest

neighbor (knn)) and selected the best method based on areas under the ROC (AUROC) across all pairwise differential abundance analyses (new **supplementary Fig. 4d,k and supplementary Fig. 9c**). Across all quantification methods (MaxQuant, apQuant, MSFragger) and all data sets, bpca resulted in overall best results and was used for comparison against IceR. In summary, missing value imputation using bpca could increase true positive rates (TPR), however, at the cost of increased false positive rates (FPR) (**Fig. 5d, Fig. 1e, Fig. 2d**). Still, IceR resulted in highest TPR (**Fig. 1e, Fig. 2d, Fig 5d**) and highest absolute true positive counts (**Fig. 1f, Fig. 2e, Fig. 5e**) while maintaining lowest FPR, especially when performing differential expression analysis on peptide-level using PECA (**Suppl Fig 9f-i**).

5. The IceR approach itself includes a total of 13 sequential steps. It is not clear, what is the importance of each step and what are the most critical steps in terms of the performance? Are all the steps really needed? Moreover, there seem to be a lot of parameters involved in the different steps. What is their role in the performance? How much parameter tuning is required for IceR and what parameters were used in the present analyses? How about MaxQuant?

Many thanks for this comment. The reason for showing these steps is to be as transparent and detailed as possible when describing the workflow of IceR instead of presenting it as a black box. Thereby, displaying this in 13 steps is meant to provide the interested reader with a rationale of the principles that are at the basis of IceR, and indicate how they are connected. This is only a visual representation to provide an understanding what is happening 'under the hood', accompanied by a supplementary text describing these steps in some detail, however most definitely this does NOT mean that the user needs to go through these steps successively and adjust settings. Instead, the user chooses a limited set of parameters (or can use default settings, see below) at the beginning of the analysis, after which IceR proceeds through all of the steps by itself.

A strong asset of IceR is that several parameters are estimated automatically based on properties of the data, such as the absolute feature alignment windows for feature-based PIP (RT- and m/z-window). Thereby this circumvents asking input from the user who usually does not have a priori knowledge of variability between samples and its impact on optimal window sizes.

IceR requires only 4 user inputs:

- 1.) Minimal RT-window size (default: 1 min) for feature-based PIP. If the automatically estimated RT-window size falls below this value, the minimal RT-window size will be used. This parameter is used to avoid determination of very small RT-windows that otherwise would limit feature-based PIP.
- 2.) Minimal m/z-window size (default: 0.001 Da) for feature-based PIP. If the automatically estimated m/z-window size falls below this value, the minimal m/z-window size will be used. This parameter is used to avoid determination of very small m/z-windows that otherwise would limit feature-based PIP.
- 3.) Minimal resolution (default: 50 grid points per dimension, e.g. RT and m/z) for kernel density estimation (KDE) maps. Higher resolution improves accuracy of peak localizations, however, comes at the cost of quadratically increasing computational workload. In our hands, a resolution of 50 proved to be a good compromise between peak localization accuracy and computational workload and should generally work well for all types of proteomics analyses. Still, IceR checks for every feature if the defined resolution exceeds the required resolution to distinguish two peaks in e.g. RT and m/z dimension. Otherwise, IceR automatically adjusts KDE resolution. This could be important in the future when mass spectrometers allow even higher m/z resolution and/or LC systems allow even narrower peak widths.

- 4.) Number of threads to be used by IceR (default: 8) for parallelization of tasks. This parameter has no effect on the final results but only on the required run time of IceR and depends on available CPU cores of the machine.

By default, all relevant IceR parameter are automatically estimated. Parameters 1 – 3 are only used to define the lower boundaries of feature-based PIP and resolution of kernel density estimations. Importantly, we like to emphasize that default settings were used for the analysis of all described data sets throughout our manuscript, and thereby allowed robust performance of IceR across a wide range of experimental and MS-related parameters:

- Various types of mass spectrometers: LTQ-Orbitrap Velos (Fig. 1), Q-Exactive (supplementary Fig. 2-3), Orbitrap Fusion (Fig. 2), Q-Exactive HF (Fig. 3-6), Orbitrap Fusion Tribrid (Fig. 7), timsTOF Pro (Fig. 5)
- Different LC gradient lengths: 20 min (Fig. 6), 60 min (Fig. 3), 70 min (Fig. 7), 85 min (Fig. 3), 110 min (supplementary Fig. 2-3), 115 min (Fig. 1), 120 min (Fig. 3-5), 180 min (Fig. 2)
- Different sample types: clinical plasma samples (Fig. 6), bulk spiked cell line lysates (e.g. Fig. 1-5), single cell lysates (Fig. 7)

Across all these highly diverse scenarios, IceR always outperformed all other tested methods which indicates that automatic parameter selection by IceR works well.

In case of MaxQuant, all used parameters are described in the materials and methods section and unless otherwise stated, we used default parameters.

6. I see the IceR software package implementation as the main contribution of this work. Currently, however, the packages falls short of its promise (IceR 1.0.0, GitHub). Despite having a Shiny interface, the package appears to rely on Windows specific features (e.g. progress bar and folder selection) thus greatly limiting its usage, for example, in HPC environments, where Linux is often favoured. This means that the statement "tested on Windows 10 but should work on other OS as well" falls short. The authors should thoroughly test their package on all operating systems. The IceR package could also benefit from being included in a public and commonly used repository, such as CRAN or Bioconductor. Even if not, the software package should be polished and pass the appropriate checks (R CMD check, or equivalent) without errors or warnings. For instance, currently, the software relies on dozens of other R packages and there are numerous warnings related to IceR replacing existing functions, which leads to potential issues when the software is used.

We thank the reviewer for this valid comment. Indeed, we should have paid more attention on the warnings during package build as well as usage of windows-specific functions which prevented usage of IceR by a broader community. These issues have now been fixed with the latest update of IceR, and it now passes R CMD check without errors and warnings, was successfully tested not only on Win10 but also on a linux distribution (Ubuntu 20.04.1) and macOS (Big Sur 11.1.0), and was even further successfully evaluated using Travis CI (<https://travis-ci.org/mathiasalksdorf/IceR>). At the moment it is intended to keep the IceR package on GitHub as it provides:

- Platform for bug reports
- Platform for questions and issues
- Seamless integration with RStudio enabling fast and efficient correction of code based on reported bugs/issues and addition of potential new functionalities

As soon as we reach a version of IceR which was comprehensively evaluated and used by a larger community, we will move IceR to CRAN.

Minor comments:

The authors could have provided more information for the reader to understand the rationale of their choices.

We are not sure what exactly the reviewer refers to. We hope that our answers to the other questions of this reviewer have provided the specific explanation(s) that (s)he is looking for.

The ROC curves in each dataset always combine the results from all the pairwise comparisons. It would be interesting to see the results separately for each comparison. More details would also be needed regarding the treatment of missing values in the comparisons. For statistical analysis, did you allow any missing values?

We now added dedicated supplementary Figures to allow investigation of ROC curves for every individual pairwise comparison per data set (**new supplementary Figs. 5, 6, and 10**). Furthermore, we added visualizations showing numbers of detected true positives and false positives per data set (**new supplementary Fig. 4f,g,m,n, supplementary Fig. 9f-i**).

Regarding missing values during differential abundance testing: missing values were allowed as long as the respective protein was at least quantified in 2 samples of each group per pairwise comparison. We added this information to the Materials and Methods section:

In case of limma and peca, missing values were allowed as long as the respective protein was at least quantified in 2 samples of each group per pairwise comparison.

The use of the term "sequencing" is a bit confusing.

We exchanged "sequencing" by "peptide sequence" where applicable.

On page 2, lines 95-96, the authors say that "It [IceR] can be seamlessly integrated with the MaxQuant suite, or with any other pre-processed label-free proteomics data set for which detected features are reported". However, the IceR software expects MaxQuant output and all benchmarks in the manuscript are carried out only for MaxQuant output. While MaxQuant is a popular non-commercial software, it would be valuable to combine IceR with other software, such as OpenMS for the construction of more automated workflows, which would increase its value and usability.

Many thanks for this comment. At the moment the IceR workflow is optimized for MaxQuant analysis outputs as this tool is broadly used by the proteomics community. Support for additional tools is anticipated for the future, however, as it would require a large effort to make IceR compatible with analysis results of additional tools, we see this currently not to be within the scope of this manuscript. Hence, we removed "or with any other pre-processed label-free proteomics data set for which detected features are reported" from the manuscript.

REVIEWER COMMENTS

Reviewer #1 (Remarks to the Author):

The authors have gone to great lengths to address my previous comments, and the results of those additional analyses are very convincing. I am very pleased that they have now included comparisons with recent state-of-the-art methods such as apQuant and MSFragger, and continue to demonstrate superior performance. My congratulations, as this truly demonstrates the performance gains that can be had when using IceR. With the added details on the hybrid PIP algorithm, it has also become much more clear exactly how the feature alignment is conducted. The detailed explanations regarding the algorithm in their response was also very useful as it makes the logic used therein very clear, thank you.

Related to the code execution issues experienced in the first iteration, I am happy to report that now indeed the software package executes much more efficiently and we were able to run in-house verifications of the claims made in the manuscript. Thus, the claim that IceR integrates efficiently with MaxQuant is certainly true in its current state.

Regarding my previous concern raised about the ability to process FAIMS data, I thank the authors for their further clarification. I understand and agree with their statement that full compatibility falls outside the scope of their work, however, I would encourage enabling the possibility of utilizing multi-CV data one way or another for widespread adaptation in the low input / single cell proteomics field. On our end, multi-CV FAIMS is still troublesome, as when manually extracting the .raw files (e.g. 2 CVs) and running the resulting files (2 mzXML files per raw file) through MaxQuant, the result files are of course specific to each single CV. We seem to be unable to get IceR to use those exact .mzXML files, and instead, IceR converts the .raw files to .mzXML, but ignoring the multi-CV nature of those files. Of course then matching back to the MaxQuant results (based on 2 result files per .raw file) is impossible, thereby rendering these data non-functional.

I by no means think that this should preclude publication of the work, but I see it as a significant hurdle for widespread adaptation of IceR in the low-input proteomics community, where FAIMS is having a huge impact on Thermo Fisher instrumentation. It would be fantastic if the authors could somehow incorporate the use of multi-CV data in an imminent update to the code after publication. Indeed, an update to the MaxQuant software supporting such data would be an optimal way to implement that too.

In its current form, I deem the manuscript suitable for publication as it showcases very elegantly the power of IceR and the authors have addressed all my concerns.

Reviewer #2 (Remarks to the Author):

The authors have addressed some of my concerns by clarifying the text, by adding new comparison results to the manuscript and by improving the software implementation. However, many of my original concerns remain.

My impression is that the work is essentially a mix or refinement of previously introduced methods. The fact that the existing implementations of the methods are not well maintained or not easy to use, makes such reimplementations (with additional incremental improvements) a worthy goal, especially if the methods have clear benefits over more recent methods or tools. While the manuscript describes various technical aspects of IceR, a question remains, how/if these technical aspects really turn into quantifiable performance benefits over existing methods or tools. IceR definitely takes its place among similar tools, but an open question is if it is a distinctive one.

An essential question is: Does IceR offer a clear benefit over the existing methods? When looking at the new results of the comparisons to other tools, I am not sure. Indeed, proper imputation seems

really to improve MaxQuant. For instance, although IceR seems to deliver more true positives especially at low concentrations, it also finds more false positives in some of the datasets when compared to MaxQuant with bpca imputation (Figure 1d, Supplementary Figure 4f-g).

Related to the comparisons, the interpretations proposed by the authors in the manuscript and in the response are not fully supported by the results. For instance, the authors state in the manuscript: "The receiver operating characteristic (ROC) showed superior performance of IceR over all 36 pairwise DE analyses on protein- and on peptide-level compared to all other tools (Fig. 1d, Supplementary Fig. 4f-g, Supplementary Fig. 5)." However, Figure 1d suggests that the best AUROC of 0.90 was obtained using MaxQuant+bpca, while the best AUROC with IceR was 0.87. In the response, the authors also say that "In brief, for all datasets, IceR resulted in best sensitivity at lowest false discovery rates especially when performing differential abundance testing on peptide-level using peca." However, the authors do not any more show the false discovery rates but, instead, have changed to false positive rates (FPR). To make such conclusions, the false discovery rates should be included.

Another missing information in the revised manuscript seems to be the total number of proteins in each comparison. That would be relevant for understanding the AUROC and FPR performance. While the number of spike-in proteins is given, the number of background proteins (or the total number of proteins) is not. If the numbers differ greatly with different methods, this could in principle affect the AUROC and FPR calculations by favouring methods with longer lists of true negatives, if the true positives have been identified similarly.

Overall, the comparisons of IceR to other methods were now more comprehensive, but I still felt them being quite non-systematic. For example, why the imputation-based approaches are missing from Figure 4? And why the detailed comparisons of the different imputation methods are not shown for all the datasets, although it is mentioned in the response that "Across all quantification methods (MaxQuant, apQuant, MSFragger) and all data sets, bpca resulted in overall best results and was used for comparison against IceR." It would be important to always show all the results systematically.

As the authors suspect themselves, MaxLFQ may hinder MaxQuant's performance with low abundance proteins, covering many of the examined comparisons in the tested spike-in and mixture datasets. Therefore, it would have been interesting to see, whether possibly using IBAQ instead of LFQ quantification in these datasets would have boosted the performance of MaxQuant and MaxQuant+bpca further, as I already requested in my previous comments. It is not clear, why the authors chose not to test this option.

Finally, when looking at the comparisons, I was very confused to notice that many of the numbers had now changed from the original version even for the methods that were already in the original comparisons, including IceR. For instance, the numbers in the original Figure 1e are not the same as in the revised Figure 1f. Similarly, for instance, the numbers in the original Figure 2d are not the same as in the revised Figure 2e. Moreover, for instance, the AUROC value of IceR in the QE-HF data has increased from 0.78 to as high as 0.95. What has changed? Such major changes to the results warrant a detailed explanation.

When it comes to the implementation, adding support for Linux is a very positive development. With the initial installation and run of the tool on Linux (latest Ubuntu 21.04), however, file selection did not work, but there was only a grayed screen. When investigating this further, the IceR manual instructed to run R as root. This does not sound right as running programs should work for normal users. There should be no need to use admin/root for any other than maintenance tasks. Why exactly IceR requires R to be run with root? Is that the reason why file selection did not work?

When trying to use the IceR package on a Windows 10 platform and the provided example dataset, the first try got stuck in the beginning with the message: "Convert raw files to mzXML". The IceR GitHub repository manual mentioned ProteoWizard as an optional installation, but it seemed to be required, as it fixed the problem. Then, after the conversion step, the application crashed with a message: "Warning: Error in makePSOCKcluster: Cluster setup failed. 5 of 5 workers failed to

connect.” This was fixed by updating RStudio to the latest version. As the authors claim to provide an easy-to-use R package, it is crucial that the documentation is accurate.

Overall, it seems that the IceR software still needs more development. I would also suggest more testing and modifying the documentation as necessary. This is particularly important, as the authors’ rationale for providing a new tool is to replace the similar previous tools that are cumbersome to use. The current implementation of IceR does not seem to provide such a solution. nd usability.

Reviewer #1 (Remarks to the Author):

The authors have gone to great lengths to address my previous comments, and the results of those additional analyses are very convincing. I am very pleased that they have now included comparisons with recent state-of-the-art methods such as apQuant and MSFragger, and continue to demonstrate superior performance. My congratulations, as this truly demonstrates the performance gains that can be had when using IceR. With the added details on the hybrid PIP algorithm, it has also become much more clear exactly how the feature alignment is conducted. The detailed explanations regarding the algorithm in their response was also very useful as it makes the logic used therein very clear, thank you.

Related to the code execution issues experienced in the first iteration, I am happy to report that now indeed the software package executes much more efficiently and we were able to run in-house verifications of the claims made in the manuscript. Thus, the claim that IceR integrates efficiently with MaxQuant is certainly true in its current state.

Regarding my previous concern raised about the ability to process FAIMS data, I thank the authors for their further clarification. I understand and agree with their statement that full compatibility falls outside the scope of their work, however, I would encourage enabling the possibility of utilizing multi-CV data one way or another for widespread adaptation in the low input / single cell proteomics field. On our end, multi-CV FAIMS is still troublesome, as when manually extracting the .raw files (e.g. 2 CVs) and running the resulting files (2 mzXML files per raw file) through MaxQuant, the result files are of course specific to each single CV. We seem to be unable to get IceR to use those exact .mzXML files, and instead, IceR converts the .raw files to .mzXML, but ignoring the multi-CV nature of those files. Of course then matching back to the MaxQuant results (based on 2 result files per .raw file) is impossible, thereby rendering these data non-functional.

I by no means think that this should preclude publication of the work, but I see it as a significant hurdle for widespread adaptation of IceR in the low-input proteomics community, where FAIMS is having a huge impact on Thermo Fisher instrumentation. It would be fantastic if the authors could somehow incorporate the use of multi-CV data in an imminent update to the code after publication. Indeed, an update to the MaxQuant software supporting such data would be an optimal way to implement that too.

In its current form, I deem the manuscript suitable for publication as it showcases very elegantly the power of IceR and the authors have addressed all my concerns.

We thank the reviewer for his/her enthusiastic support of the manuscript.

Reviewer #2 (Remarks to the Author):

The authors have addressed some of my concerns by clarifying the text, by adding new comparison results to the manuscript and by improving the software implementation. However, many of my original concerns remain.

My impression is that the work is essentially a mix or refinement of previously introduced methods. The fact that the existing implementations of the methods are not well maintained or not easy to use, makes such reimplementations (with additional incremental improvements) a worthy goal,

especially if the methods have clear benefits over more recent methods or tools. While the manuscript describes various technical aspects of IceR, a question remains, how/if these technical aspects really turn into quantifiable performance benefits over existing methods or tools. IceR definitely takes its place among similar tools, but an open question is if it is a distinctive one.

An essential question is: Does IceR offer a clear benefit over the existing methods? When looking at the new results of the comparisons to other tools, I am not sure. Indeed, proper imputation seems really to improve MaxQuant. For instance, although IceR seems to deliver more true positives especially at low concentrations, it also finds more false positives in some of the datasets when compared to MaxQuant with bPCA imputation (Figure 1d, Supplementary Figure 4f-g).

The reviewer has the impression that IceR is a 'refinement of previously introduced methods', which is true to the extent that we did not invent direct ion current extraction. However, we like to emphasize that DICE is only rarely used in proteomics, where instead most people resort to imputation. Moreover, we implemented DICE in a hybrid manner with feature-based peptide identity propagation, which is unique to IceR offering distinct advantages not present in previous tools. Moreover, we like to stress that, simultaneously, we incorporated various other innovations that are unique to IceR, most notably the advanced alignment strategy, and the sample- and feature-specific alignments based on kernel density estimated ion accumulation maps. To integrate all these functionalities and improvements, we designed IceR from scratch in a modern unified algorithm, not only providing a streamlined alternative for previous approaches employing DICE, but also adding new and powerful functionality.

While we introduce IceR in Figure 1A, we use all other figures in the manuscript to assess superior performance metrics compared to other approaches (Figures 1B-F, 2, 3, 4 and 5, backed up by a string of supplementary figures). Moreover, we demonstrate the upshot of these improvements in real-life experiments (Figs 6 and 7), retrieving biological information that had remained hidden when using MaxQuant with imputation in the publications where these data were retrieved from. Perhaps these latter two figures, the first in global proteomics and the second in a single-cell application, best illustrate the distinctive advantage of IceR that this reviewer is looking for. Hence, instead of making assumptions which imputation method may be best for their experiment, with IceR scientists have an alternative to directly extract quantitative information from low-level (yet measured) intensities in their experimental data, with superior performance.

If we are allowed to draw an analogy: the first electrical car was introduced in the 1870s, however this was quickly and completely superseded by combustion-driven vehicles. Of course, one can then claim that a Tesla is a 'refinement of previously introduced methods', however this has not come without the innovations that effectively solved shortcomings of now traditional cars.

We like to add another comment on the perceived equivalence of MaxQuant with imputation over IceR, related to the specific example the reviewer refers to (Figure 1d, Supplementary Figure 4f-g). It is true that for this particular data set IceR showed a slightly higher false positive rate compared to MaxQuant (~0.5 % vs 0.15%). However, IceR showed by far the best true positive rate (67%) compared to all other approaches including MaxQuant without (20%) and with bPCA imputation (55%). Furthermore, in all other data sets IceR outperformed all other approaches even more clearly (Fig. 2, Fig. 4, Fig. 5), overall showing the clear performance benefit of IceR compared to all other approaches including additional imputation. In addition, as evidenced by all data sets, missing value imputation can show variable performance depending on the used quantification workflow (e.g. MaxQuant or MSFragger or apQuant) and on the data itself. Hence, the best imputation method might not always be only a single approach (e.g. shown here <https://doi.org/10.1093/bib/bbx054>). However, how would one select the best imputation approach in a real data set where the ground truth is unknown? This problem is circumvented by IceR as the user does not have to make any decision. As shown across all

data sets, IceR will perform at least similarly well as the best imputation method as it uses real observed signal intensities instead of artificially introduced data.

Related to the comparisons, the interpretations proposed by the authors in the manuscript and in the response are not fully supported by the results. For instance, the authors state in the manuscript: “The receiver operating characteristic (ROC) showed superior performance of IceR over all 36 pairwise DE analyses on protein- and on peptide-level compared to all other tools (Fig. 1d, Supplementary Fig. 4f-g, Supplementary Fig. 5).” However, Figure 1d suggests that the best AUROC of 0.90 was obtained using MaxQuant+bpca, while the best AUROC with IceR was 0.87.

Many thanks for this comment. The AUROC is generally a good measure to compare the capability of methods to rank and distinguish true from false hits for every possible cut-off. However, typically cut-off criteria are set to 1% or 5% false positive rate (i.e. 99% or 95% specificity). Hence, it is even more important for a method to achieve a high sensitivity at the applied specificity cut-off. When looking at the ROC curve (Fig. 1d) and applying a 95 % specificity cut-off, IceR results in a higher sensitivity (72.5%) compared to MaxQuant without (64%) and with bpca imputation (70%). Nonetheless, our wording was unclear and we adjusted the sentence to:

“The receiver operating characteristic (ROC) showed superior performance of IceR over all 36 pairwise DE analyses on protein- and on peptide-level compared to all other tools e.g. when applying a specificity cut-off at 5 % (Fig. 1d, Supplementary Fig. 4f-g, Supplementary Fig. 5).”

We also revised Fig. 1d, Fig. 2c, Fig. 4b,e, and Fig. 5b,c to improve readability of sensitivities at a 95 % specificity cut-off.

In the response, the authors also say that “In brief, for all datasets, IceR resulted in best sensitivity at lowest false discovery rates especially when performing differential abundance testing on peptide-level using peca.” However, the authors do not any more show the false discovery rates but, instead, have changed to false positive rates (FPR). To make such conclusions, the false discovery rates should be included.

Many thanks for spotting this unclear statement. We meant to indicate false positive rates instead of false discovery rates. Therefore, we correct our statement to:

“In brief, for all datasets, IceR resulted in best sensitivity at lowest false positive rates especially when performing differential abundance testing on peptide-level using peca.”

Another missing information in the revised manuscript seems to be the total number of proteins in each comparison. That would be relevant for understanding the AUROC and FPR performance. While the number of spike-in proteins is given, the number of background proteins (or the total number of proteins) is not. If the numbers differ greatly with different methods, this could in principle affect the AUROC and FPR calculations by favouring methods with longer lists of true negatives, if the true positives have been identified similarly.

The numbers of background proteins for the respective results in Fig. 1, 2, 4, and 5 are given in supplementary Fig. 4a,h, supplementary Fig. 8a,g, and in Fig. 5a, respectively. In general, numbers are comparable and IceR shows only slightly higher numbers of quantified background proteins. Hence, AUROC and FPR calculations should not be affected.

Overall, the comparisons of IceR to other methods were now more comprehensive, but I still felt them being quite non-systematic. For example, why the imputation-based approaches are missing from Figure 4?

Many thanks for spotting this inconsistency. We now added results for MaxQuant with bpca imputation (bpca worked again best compared to all other imputation methods as shown in supplementary Fig. 8d,j). Similar to previous observations, imputation of missing values generally improved true positive detection, but can also result in highly increased false positive rates (Fig. 4f). IceR again showed superior performance.

And why the detailed comparisons of the different imputation methods are not shown for all the datasets, although it is mentioned in the response that “Across all quantification methods (MaxQuant, apQuant, MSFragger) and all data sets, bpca resulted in overall best results and was used for comparison against IceR.” It would be important to always show all the results systematically.

The performance of the tested imputation methods across all tools are shown in the corresponding supplementary figures (supplementary Fig. 4 d,k, supplementary Fig. 8d,j, and supplementary Fig. 9c). Furthermore, the focus of our work was on the implementation and validation of IceR, not the detailed evaluation of imputation methods.

As the authors suspect themselves, MaxLFQ may hinder MaxQuant’s performance with low abundance proteins, covering many of the examined comparisons in the tested spike-in and mixture datasets. Therefore, it would have been interesting to see, whether possibly using iBAQ instead of LFQ quantification in these datasets would have boosted the performance of MaxQuant and MaxQuant+bpca further, as I already requested in my previous comments. It is not clear, why the authors chose not to test this option.

As already indicated in our previous reply, there are several reasons why IceR should be compared to MaxQuant with LFQ instead of iBAQ intensities:

- 1.) MaxLFQ-based quantification results overall in most accurate and reliable quantifications (DOI: 10.1016/j.jprot.2020.103669)
- 2.) IceR itself implements the MaxLFQ algorithm for estimation of protein abundance levels from peptide intensities
- 3.) Most importantly, the MaxLFQ algorithm is designed to compare protein abundance BETWEEN samples, while iBAQ is designed to do this WITHIN a sample (Schwanhäusser et al, Nature 2011). Critically, iBAQ lacks the outlier removal and normalization procedures that are implemented in MaxLFQ to allow inter-sample comparisons. Hence, using iBAQ values for the comparisons across samples as suggested by the reviewer would make improper use of the iBAQ algorithm, and therefore we have stayed away from this.

Finally, when looking at the comparisons, I was very confused to notice that many of the numbers had now changed from the original version even for the methods that were already in the original comparisons, including IceR. For instance, the numbers in the original Figure 1e are not the same as in the revised Figure 1f. Similarly, for instance, the numbers in the original Figure 2d are not the same as in the revised Figure 2e. Moreover, for instance, the AUROC value of IceR in the QE-HF data has increased from 0.78 to as high as 0.95. What has changed? Such major changes to the results warrant a detailed explanation.

We apologize for the confusion that was raised, and we are happy to explain the reasons for the slight changes in numbers:

Original Fig. 1e showed true positive rates and false positive rates on peptide level, while the revised figure (now Fig. 1f) now shows TPR and FPR on protein level. We did this since the other method that compared best to IceR (MaxQuant with bpca imputation) showed best results on protein-level. We apologize for not indicating this change in our previous rebuttal.

The other changes in numbers are explained by the fact that all analyses were re-run completely (including the DE analyses using limma). The reason for this is that between the first submission and the resubmission, versions of both R (3.6.3 to 4.0.2) and limma (3.42.2 to 3.44.3) underwent an update. This had some minor effect on estimated fold-changes (second digit after the decimal point) and t-values (first digit after the decimal point). The changelog between the limma versions indicates several optimizations/corrections of the contrasts.fit() function for cases with missing values in the expression matrix. This function is used to extract the differential testing results for the respective contrasts of interest. The updates in this function most likely explain the minor deviations in numbers between both of our submissions. However, since we have reprocessed all data sets, these changes affected the results of all tools/conditions shown in the manuscript and do not affect any of our conclusions. Also here, we apologize for not indicating this change in our previous rebuttal.

In case of the AUROCs indicated in Fig. 5, the original ROC analyses were performed for the data from QE-HF and timsTOF Pro together. For a better comparison of the performances of the respective tools for the revised figure we decided to split the analyses and evaluate the QE-HF and timsTOF Pro data separately.

When it comes to the implementation, adding support for Linux is a very positive development. With the initial installation and run of the tool on Linux (latest Ubuntu 21.04), however, file selection did not work, but there was only a grayed screen. When investigating this further, the IceR manual instructed to run R as root. This does not sound right as running programs should work for normal users. There should be no need to use admin/root for any other than maintenance tasks. Why exactly IceR requires R to be run with root? Is that the reason why file selection did not work?

As indicated in the IceR instructions, root privileges are only required during installation (in our hands root was only required for install_github() from devtools). As also described in the instructions, no root is required for the usage of IceR. Prompted by the reviewer's comment, we again tested IceR for Ubuntu on a dedicated and a virtual PC by 2 independent people with fresh installations of R and could not reproduce the issues. Were all required essential libraries installed as described in the instructions?

When trying to use the IceR package on a Windows 10 platform and the provided example dataset, the first try got stuck in the beginning with the message: "Convert raw files to mzXML". The IceR GitHub repository manual mentioned ProteoWizard as an optional installation, but it seemed to be required, as it fixed the problem. Then, after the conversion step, the application crashed with a message: "Warning: Error in makePSOCKcluster: Cluster setup failed. 5 of 5 workers failed to connect." This was fixed by updating RStudio to the latest version. As the authors claim to provide an easy-to-use R package, it is crucial that the documentation is accurate.

Installation of ProteoWizard is optional. However, as stated in the instructions, if it is not installed, the user has to prepare mzXML files manually and store them in a required folder. Hence, for a simpler usage of IceR it is recommended to install ProteoWizard but the user can decide. Furthermore, as clearly indicated in the manual on Github, for running the example data set in the way as described, installation of ProteoWizard is required: "For this example, we will require ProteoWizard to be installed as raw files will have to be converted". Further, we could not reproduce the observed makePSOCKcluster error and it should also be independent from the used RStudio version. Was the right R version (> 3.63 on Windows) selected in RStudio before upgrading RStudio? Were the important notes in the section "Prerequisites" considered (blank spaces and special letters in file paths)?

Overall, it seems that the IceR software still needs more development. I would also suggest more testing and modifying the documentation as necessary. This is particularly important, as the authors' rationale for providing a new tool is to replace the similar previous tools that are cumbersome to use. The current implementation of IceR does not seem to provide such a solution.

We are sorry to hear about the issues that this reviewer experienced getting IceR to run, but we were happy to note that updating to the latest version of R, as indicated in the IceR instructions, solved the issue. In addition, Reviewer 1 did not experience these problems, who confirmed that 'the claim that IceR integrates efficiently with MaxQuant is certainly true in its current state'. We are fully aware that additional functionality is desirable, and therefore we appreciate both reviewers' suggestions as a confirmation that IceR in its present form is a powerful tool for sensitive proteome analysis, and we certainly interpret it as an encouragement to extend IceR in the future.